# Improving Planning with Large Language Models: A Modular Agentic Architecture

## Abstract

Large language models (LLMs) demonstrate impressive performance on a wide variety of tasks, but they often struggle with tasks that require multi-step reasoning or goal-directed planning. Both cognitive neuroscience and reinforcement learning (RL) have proposed a number of interacting functional components that together implement search and evaluation in multi-step decision making. These components include conflict monitoring, state prediction, state evaluation, task decomposition, and orchestration. To improve planning with LLMs, we propose an agentic architecture, the Modular Agentic Planner (MAP), in which planning is accomplished via the recurrent interaction of the specialized modules mentioned above, each implemented using an LLM. MAP improves planning through the interaction of specialized modules that break down a larger problem into multiple brief automated calls to the LLM. We evaluate MAP on three challenging planning tasks – graph traversal, Tower of Hanoi, and the PlanBench benchmark – as well as an NLP task requiring multi-step reasoning (strategyQA). We find that MAP yields significant improvements over both standard LLM methods (zero-shot prompting, in-context learning) and competitive baselines (chain-of-thought, multi-agent debate, and tree-of-thought), can be effectively combined with smaller and more cost-efficient LLMs (Llama3-70B), and displays superior transfer across tasks. These results suggest the benefit of a modular and multi-agent approach to planning with LLMs.

## 1 Introduction

Large Language Models (LLMs) (Devlin et al., 2019; Brown et al., 2020) have become widely accepted as highly capable generalist systems with a surprising range of emergent capacities (Srivastava et al., 2022; Wei et al., 2022a; Webb et al., 2023). They have also sparked broad controversy, with some suggesting that they are approaching general intelligence (Bubeck et al., 2023), and others noting a number of significant deficiencies (Mahowald et al., 2023). A particularly notable shortcoming is their poor ability to plan or perform faithful multi-step reasoning (Valmeekam et al., 2023; Dziri et al., 2023). Recent work (Momennejad et al., 2023) has evaluated the extent to which LLMs might possess an emergent capacity for planning and exploiting *cognitive maps*, the relational structures that humans and other animals utilize to perform planning (Tolman, 1948; Tavares et al., 2015; Behrens et al., 2018). This work found that LLMs displayed systematic shortcomings in planning tasks that suggested an inability to reason about cognitive maps. Common failure modes included a tendency to 'hallucinate' (e.g., to use non-existent transitions and paths), and to fall into loops. This work raises the question of how LLMs might be improved so as to enable a capacity for planning, especially given the ubiquity of sequential decision making, reasoning, and planning problems across the wide application of generative AI and LLMs.

Here, we take a step toward improving planning with LLMs, by taking inspiration from both cognitive neuroscience and formal theories of decision-making and planning. In traditional theories of planning, such as those found in the field of reinforcement learning (RL) (Sutton & Barto, 2018), planning is carried out via the interaction of several specialized functions or modules, rather than through the activity of a single, monolithic system. For instance, many approaches involve distinct functions for action proposal, state evaluation, subgoal identification, or state prediction, many of which have also been related to the function of specific brain regions (see Section 6 for discussion). An interesting observation is that LLMs are often able to carry out these functions when probed

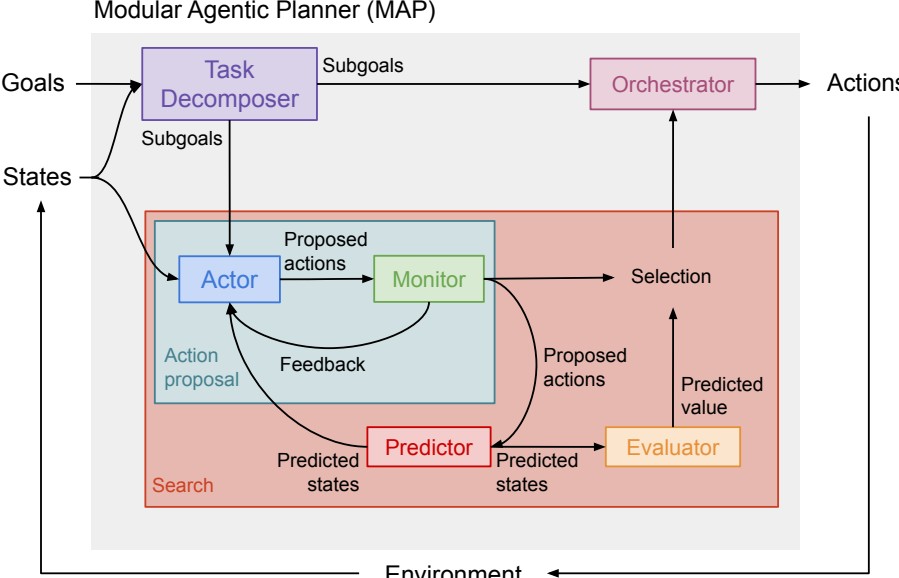

Figure 1: **Modular Agentic Planner (MAP).** The agent receives states from the environment and high-level goals. These are processed by a set of specialized LLM modules. The TaskDecomposer receives high-level goals and generates a series of subgoals. The Actor generates proposed actions given a state and a subgoal. The Monitor gates these proposed actions based on whether they violate certain constraints (e.g., task rules) and provides feedback to the Actor. The Predictor predicts the next state given the current state and a proposed action. The Evaluator is used to estimate the value of a predicted state. The Predictor and Evaluator are used together to perform tree search. The Orchestrator determines when each subgoal has been achieved, and when the final goal has been achieved, at which point the plan is emitted to the environment as a series of actions.

in isolation, but are unable to reliably integrate and orchestrate these capacities in the service of a goal. For instance, Momennejad et al. (2023) noted that LLMs often attempt to traverse invalid or hallucinated paths in planning problems (e.g., to move between rooms that are not connected), even though they can correctly identify these paths as invalid when probed separately. This suggests the possibility of an agentic approach using LLMs, in which planning is carried out through the coordinated and recurrent interaction of multiple LLM modules, each of which is specialized to perform a distinct process.

With this goal in mind, we propose the Modular Agentic Planner (MAP) (Figure 1), an agentic architecture composed of modules that are specialized to perform specific functions within the planning process. Specifically, we have identified and implemented the following key modules: error monitoring, action proposal, state prediction, state evaluation, task decomposition, and task coordination. Action proposal, state prediction, and state evaluation are further combined to perform tree search. All modules are implemented using an LLM, which receives instructions describing the module's role via prompting and few-shot in-context learning (ICL). The resulting MAP algorithm (Algorithm 2) is implemented via the recurrent interaction of these modules, combining the strengths of classical planning and search algorithms with the use of LLMs as general-purpose world models and planning functions.

We evaluate MAP on four challenging decision-making tasks that require planning and multi-step reasoning. First, we performed controlled experiments on a set of graph traversal tasks according to the CogEval protocol (Momennejad et al., 2023). These tasks require goal-directed navigation in novel environments (MDPs) described in natural language, of which we selected an environment that was most challenging for LLMs, including GPT-4. Second, we investigate Tower of Hanoi (ToH), a classic problem solving task that requires multi-step planning (Simon, 1975). Third, we investigate the two most challenging tasks in the PlanBench benchmark: mystery BlocksWorld and

Logistics (Valmeekam et al., 2023). Finally, we investigate a challenging NLP task that requires multi-step reasoning, StrategyQA (Geva et al., 2021). We find that, when implemented with GPT-4, MAP significantly improves performance on all four tasks (Figures 2 and 3, Tables 1 and 2), and that the approach can also be effectively implemented with a smaller and more cost-efficient LLM (Llama3-70B, Table 10). Transfer experiments further indicate that MAP displays an improved ability to generalize between tasks, and ablation experiments indicate that each of the individual modules plays an important role in the overall architecture's performance (Figure 3). Taken together, these results indicate the potential of a modular agentic approach to improve the reasoning and planning capabilities of LLMs.

## 2 APPROACH

### 2.1 PROBLEM FORMULATION

We consider planning problems in environments that are both deterministic and fully observable. The environment and transition dependencies are described to the agent, along with a starting state $x$ and a goal $y$, and the agent must generate a plan $P$ (a series of actions) without directly interacting with the environment. Despite the relatively limited nature of the environments we consider (fully observable, deterministic), planning problems of this sort are nevertheless extremely challenging for LLMs (Valmeekam et al., 2023; Momennejad et al., 2023).

### 2.2 MODULES

MAP contains the following specialized modules, each constructed from a separate LLM instance through a combination of prompting and few-shot ($\leq 3$ examples) in-context learning (described in greater detail in section A.7):

- **TaskDecomposer**. The TaskDecomposer receives the current state $x$ and a goal $y$ and generates a set of subgoals $Z$ that will allow the agent to gradually work toward its final goal. In the present work, the TaskDecomposer is only utilized to generate a single intermediate goal, though in future work we envision that it will be useful to generate a series of multiple subgoals.

- **Actor**. The Actor receives the current state $x$ and a subgoal $z$ and proposes $B$ potential actions $A = a_{b=1} \ldots a_{b=B}$. The Actor can also receive feedback $\epsilon$ from the Monitor about its proposed actions.

- **Monitor**. The Monitor gates the actions proposed by the Actor based on their validity (e.g., whether they violate the rules of a task). It emits an assessment of validity $\sigma$, and also feedback $\epsilon$ in the event the action is deemed invalid.

- **Predictor**. The Predictor receives the current state $x$ and a proposed action $a$ and predicts the resulting next state $\tilde{x}$.

- **Evaluator**. The Evaluator receives a next-state prediction $\tilde{x}$ and produces an estimate of its value $v$ in the context of goal $y$. This is accomplished by prompting the Evaluator (and demonstrating via a few in-context examples) to estimate the minimum number of steps required to reach the goal (or subgoal) from the current state.

- **Orchestrator**. The Orchestrator receives the current state $x$ and a subgoal $z$ and emits an assessment $\Omega$ of whether the subgoal has been achieved. When the Orchestrator determines that all subgoals (including the final goal) have been achieved, the plan is emitted to the environment as a series of actions.

### 2.3 ACTION PROPOSAL LOOP

This section describes MAP's algorithms, the first of which is the action proposal loop. The Actor and Monitor interact via the ProposeAction function (Supplementary Algorithm 1). The Actor proposes a set of potential actions, which are then gated by the Monitor. If the Monitor determines that the actions are invalid (e.g., they violate the rules of a task), feedback is provided to the Actor, which then proposes an alternative action. The output of the ProposeAction function is a set of

potential actions, one of which will be selected as the action at the next time step (as described in the following section).

---

**Algorithm 1: Action proposal loop.** ProposeAction takes a state $x$ and a goal $y$ and generates $B$ potential actions $A = a_{b=1} \ldots a_{b=B}$. This is implemented via a loop, in which the Actor first proposes potential actions, and the Monitor then assesses those actions according to certain constraints (e.g., task rules), providing feedback if any of the actions are deemed to be invalid. This continues until the proposed actions are considered valid. See Sections A.7.2 and A.7.3 for more details.

**Function** ProposeAction $(x, y, B)$**:**
    $\sigma \leftarrow$ false                                 // Initialize validity
    $E \leftarrow \{\}$                                  // Initialize feedback
    **while** $\sigma$ is false **do**
        $A \leftarrow$ Actor$(x, y, E, B)$                    // Sample B actions
        $\sigma, \epsilon \leftarrow$ Monitor$(x, A)$     // Determine validity and provide feedback
        $E \leftarrow E \cup \{\epsilon\}$                      // Accumulate feedback
    **end**
**return** $A$

---

## 2.4 TREE SEARCH

ProposeAction is further embedded in a Search loop (Supplementary Algorithm 3). The actions emitted by ProposeAction are passed to the Predictor, which predicts the states that will result from these actions. A limited tree search is then performed, starting from the current state, and then exploring $B$ branches recursively to a depth of $L$ layers. Values are assigned to the terminal states of this search by the Evaluator, and the action leading to the most valuable predicted state is selected.

---

**Algorithm 2: Modular Agentic Planner (MAP).** MAP takes a state $x$ and a goal $y$ and generates a plan $P$, a series of actions with a maximum length of $T$. The TaskDecomposer first generates a set of subgoals $Z$. The agent then pursues each individual subgoal $z$ in sequence, followed by the final goal $y$. At each time step, Search (Algorithm 3) is called to generate an action and a predicted next-state. Actions are added to the plan until the Orchestrator determines that the goal has been achieved, or the plan reaches the maximum length $T$.

**Function** MAP $(x, y, T, L, B)$**:**
    $P \leftarrow []$                                   // Initialize plan
    $Z \leftarrow$ TaskDecomposer$(x, y)$              // Generate subgoals
    **for** $g$ in $1 \ldots$ length$(Z) + 1$ **do**
        **if** $g \leq$ length$(Z)$ **then**
            $z \leftarrow Z_g$                 // Update current subgoal
        **else**
            $z \leftarrow y$                    // Final goal
        **end**
        $\Omega \leftarrow$ Orchestrator$(x, z)$       // Initialize subgoal assessment
        **while** $\Omega$ is false and length$(P) < T$ **do**
            $a, x, v \leftarrow$ Search$(l = 1, L, B, x, z)$       // Perform search
            $P \leftarrow P$.append$(a)$               // Update plan
            $\Omega \leftarrow$ Orchestrator$(x, z)$    // Determine if subgoal is achieved
        **end**
    **end**
**return** $P$

---

## 2.5 PLAN GENERATION

Algorithm 2 describes the complete MAP algorithm. To generate a plan, the TaskDecomposer component of MAP first generates a set of subgoals based on the final goal and current state. These subgoals guide the search and are internally pursued one at a time, utilizing the Search loop to generate actions until the Orchestrator determines that the subgoal has been achieved. The actions are accumulated in a plan buffer $P$ until either the Orchestrator determines that the final goal has been reached, or the maximum allowable number of actions $T$ are accumulated.

## 3 EXPERIMENTS

Experiment details are described in Section A.3. Code is available at: https://github.com/MAPLLM/MAPICLR2025sub.

### 3.1 TASKS

**Graph Traversal.** We performed experiments on four multi-step planning tasks based on graph traversal using the CogEval protocol (Momennejad et al., 2023). Natural language descriptions of a graph are provided with each node assigned to a room (e.g., 'room 4 is connected to room 7'). The tasks included Valuepath, involving finding the shortest path between a given room and the largest of two possible rewards (but without going through the room with the smaller reward); Steppath, involving finding the shortest path between two rooms; Detour, in which the Valuepath task is first described, after which an edge is subsequently removed from the graph; and Reward Revaluation, in which the Valuepath task is first described, and the value associated with two reward locations is subsequently changed. Please see Section A.4 in the Appendix for more details.

**Tower of Hanoi.** We also investigated a classic multi-step planning task called the Tower of Hanoi (ToH) (Figure 5). In the original task, there are three pegs and a set of disks of different sizes. The disks must be moved into a particular goal configuration, while observing a set of constraints that prevent simple solutions. In our experiments, we designed an alternative (but isomorphic) formulation of this task in which the inputs are text-based rather than visual. This text-based formulation made it possible to evaluate language models on the task, but it also resulted in a task that does not share any surface features with the original task, making it unlikely that GPT-4 could rely on exposure to descriptions of ToH in its training data to solve the problem. Please see Section A.4 in the Appendix for more details.

**PlanBench.** To assess the generality and robustness of our approach, we also investigated a more extensive planning benchmark, PlanBench, consisting of synthetically generated problems in a number of distinct domains. We specifically investigated the Logistics domain, involving the transportation of goods between cities using airplanes and trucks, and the Mystery Blocksworld (deceptive) domain, which involves arbitrary names for entities and actions, and is the most challenging domain in the dataset (more details can be found in Valmeekam et al. (2023)).

**StrategyQA.** Finally, to test the extent to which MAP can be applied to more real-world tasks, we investigated StrategyQA, an NLP task that requires multi-step reasoning, and has proven challenging for standard LLM methods (Geva et al., 2021). In this task, an unusual question is posed (e.g. 'Did Aristotle own a laptop?') that requires multi-step reasoning. The question must be decomposed into sub-questions which must be successively solved in order to arrive at a final answer (more details can be found in Geva et al. (2021))

### 3.2 BASELINES

We compared our model to several baseline methods. The first method involved asking GPT-4 (zero-shot) to provide the solution step by step. For the second method, in-context learning (ICL), we provided GPT-4 with a few in-context examples of a complete solution. We provided two examples for ToH, Valuepath, Detour, and Reward Reval, and three examples for Steppath (one each for 2, 3, and 4 steps) and PlanBench. The third method was chain-of-thought (CoT) (Wei et al., 2022b). For this method, the in-context examples were annotated with a series of intermediate computations that break down the planning process into multiple steps (see Sections A.7.7-A.7.9 for example baseline prompts). The fourth method was multi-agent debate (MAD), using the codebase from Du et al. (2023). In this approach, similar to MAP, a solution is generated through the interaction between multiple LLM instances (each instance was equivalent to the GPT-4 ICL baseline); however, unlike MAP, these instances are not specialized to perform specific functions. Finally, we investigated tree-of-thought (ToT), using the original codebase from Yao et al. (2023). Similar to MAP, ToT uses multiple LLM modules to perform tree search, although MAP incorporates additional modules and control processes (see Section A.2). To ensure that ToT was given the best chance of performing well on our tasks, we tested two versions, one with prompts that were similar to those in the original implementation (shown in the main results Section 4), and one that incorporated prompts from MAP (ToT-MAP, see Appendix Table 8). For ToT, multiple potential plans are generated for each

problem, and a method is required to select one of these plans (a problem-specific heuristic was used in the original work). To ensure a fair comparison, we evaluated using two metrics, 1) the best plan (according to a groundtruth evaluation) for each problem, 2) the average performance of all plans for each problem. We report both of these metrics for both ToT and MAP.

### 3.3 METRICS

Our primary metric is the percentage of problems that are solved for any given task. For non-deterministic methods (MAP, ToT), we consider both average percent solved (across multiple plans per problem) and percent solved for the best plan (according to a groundtruth evaluation). For Tower-of-Hanoi and Graph Traversal, we also consider the percentage of moves that are invalid (that violates the rules, or attempts to traverse a nonexistent edge).

## 4 RESULTS

Figure 2 shows the results on the four graph traversal tasks (see Section A.5 for all results in tabular form). On the Valuepath task, MAP solved 100% of problems, outperforming all baselines (outperforming the next-best baselines, GPT-4 CoT and ICL, by a margin of 9%). On the Steppath task, MAP displayed perfect performance for 2-step and 3-step paths, and near-perfect performance for 4-step paths, again outperforming all baselines (outperforming the next-best baseline, ToT, by a margin of 45%). MAP also outperformed all baselines on the Detour task (outperforming the next-best baseline, GPT-4 CoT, by a margin of 16%), and performed on par with the baselines on the Reward Revaluation task (while still outperforming GPT-4 zero-shot). This demonstrates that MAP can flexibly adjust to new circumstances when generating plans. Finally, the model did not propose any invalid actions in any of the four tasks (i.e., it did not hallucinate the presence of non-existent edges), due to the filtering of invalid actions by the Monitor (Figure 6).

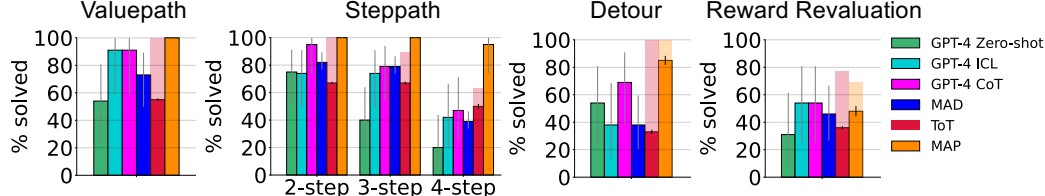

Figure 2: **Graph traversal results.** '% solved' indicates percentage of problems solved without proposing invalid actions (↑ better). GPT-4 Zero-shot, ICL, COT, and MAD baselines are deterministic, and therefore a single run was performed on all problems. Note that MAP did not employ tree search on the Steppath task, and did not employ task decomposition on any of the graph traversal tasks. Without tree search, MAP's performance is deterministic, and therefore only a single run was performed on the Steppath task, whereas we performed 5 runs with ToT. Gray error bars reflect 95% binomial confidence intervals (for models evaluated on a single run). Dots reflect values of 0%. Dark bars indicate average performance over multiple plans/runs. Light bars indicate best performance. For Valuepath, Detour, and Reward Revaluation we performed 10, 10, and 5 runs respectively with MAP and ToT, and present average performance ± the standard error of the mean (black error bars).

Figure 3 shows the results on Tower of Hanoi (ToH). MAP demonstrated a significant improvement both in terms of the number of problems solved (left) and the number of invalid actions proposed (right). On 3-disk problems, MAP yielded a nearly seven-fold improvement in the number of problems solved over zero-shot performance, and significantly outperformed standard in-context learning (ICL; by 28%), chain-of-thought (CoT; by 32%), multi-agent debate (MAD; by 49%), and tree-of-thought (ToT; by 68%). When considering the best plan (out of 5 runs) for each problem, MAP achieved a perfect score of 100%. MAP's improved performance relative to MAD demonstrates the importance of interaction between *specialized* LLM instances (i.e., a modular approach), whereas MAD involves interactions between multiple LLM instances prompted to perform the same task. MAP's superior performance relative to ToT demonstrates that tree search, though an important part of the approach, is not sufficient to explain MAP's performance, and the other modules play an

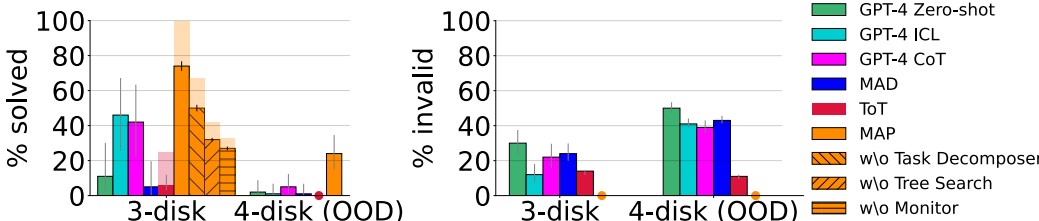

Figure 3: **Tower of Hanoi (ToH) results.** '% solved' indicates percentage of problems solved without proposing invalid actions (↑ better). '% invalid' indicates percentage of moves that are invalid (↓ better). Note that 4-disk problems are out-of-distribution (OOD). GPT-4 Zero-shot, ICL, CoT, and MAD baselines are deterministic and reflect a single run. Gray error bars reflect 95% binomial confidence intervals. Dots reflect values of 0%. Dark bars indicate average performance over multiple plans/runs. Light bars indicate best performance. MAP results for 3-disk problems reflect the average over 5 runs ± the standard error of the mean (black error bars). MAP results for 4-disk problems reflect a single run, due to the high computational cost of multiple runs.

important role. For the problems that MAP solved, the average plan length (5.4) was close to the optimal number of moves (4.4). The model also demonstrated some ability to generalize out-of-distribution (OOD) to more complex 4-disk problems (not observed in any in-context examples), whereas the baseline models solved close to 0% of these problems. Notably, MAP did not propose any invalid actions, even on OOD 4-disk problems, whereas the baselines proposed a significant number of invalid actions.

We also performed several experiments with ToH to better understand the MAP algorithm. First, we investigated how MAP's performance varied as a function of the depth of tree search. We found that a depth of $L = 2$ provided the best tradeoff between performance and cost (Table 14). A depth of $L = 1$ resulted in worse performance, while a depth of $L = 3$ incurred greater cost without significantly improving performance. Second, we investigated the extent to which MAP's performance on this task depended on being provided with an explicit strategy for task decomposition (the goal recursion strategy). To address this, we provided the GPT-4 zero-shot, ICL, and CoT baselines with a description of this strategy. We found that this strategy did improve baseline performance, but MAP still outperformed these baselines even when they were provided with the strategy (Table 12). Third, we investigated whether the computational costs of the MAP algorithm could be mitigated by using a smaller LLM. We found that a version of MAP that used Llama3-70B still outperformed baselines that used the same LLM, and even outperformed the best GPT-4 baseline, GPT-4 ICL (Table 10).

Table 1 shows the results for the PlanBench dataset, where MAP outperformed all of the baselines that we considered. Notably, due to the complexity of the problems in this dataset, it was very costly to perform tree search, so we evaluated a minimal version of MAP that did not involve tree search. For this same reason, we were unable to evaluate a ToT baseline on the full set of problems, but we include a comparison with ToT on a subset of problems in Table 13. Even without tree search, MAP outperformed ToT. These results demonstrate that, although MAP certainly benefits from the use of tree search, it can still provide significant performance benefits in domains where this is not feasible. We also investigated a zero-shot version of MAP (without in-context examples) on the most challenging PlanBench domain (mystery blocksworld), and found that it outperformed both GPT-4 zero-shot and GPT-4 ICL (Table 11), suggesting that MAP can be useful even in settings where in-context examples are not available.

Table 2 shows the results for the StrategyQA benchmark, where MAP outperformed both CoT and ToT, and performed on par with human participants. This demonstrates the potential of MAP to be beneficial in more real-world tasks, such as question-answering tasks that require multi-step reasoning.

Table 1: PlanBench results.[1]

| Model | Logistics | Mystery BW |
|---|---|---|
| GPT-4 Zero-shot | 7 | 0.2 |
| GPT-4 ICL | 12 | 7.8 |
| MAD | 16.2 | 7.3 |
| GPT-4 CoT | 17 | 10.6 |
| **MAP** | **24** | **27.4** |

Table 2: StrategyQA results. [2]

| Model | Accuracy |
|---|---|
| ToT | 81.7 [3] $\pm$ 1.2 |
| GPT-4 CoT | 84.7 $\pm$ 0.3 |
| **MAP** | **87.7 $\pm$ 0.7** |
| Human [4] | 87.0 |

Finally, we performed transfer experiments to study whether few-shot in-context learning would support generalization to different planning tasks. Table 3 shows the results for these experiments, including results for transfer from planning on a smaller graph to planning on a larger graph (n7tree $\rightarrow$ n15star), transfer to a semantically distinct but structurally isomorphic task (blocksworld (BW) $\rightarrow$ mystery blocksworld (mystery BW)), and transfer between completely different tasks (ToH $\rightarrow$ Mystery BW). We found that MAP outperformed both GPT-4 ICL and CoT in each of these settings, indicating that MAP can improve the generalizability and robustness of planning in LLMs.

Table 3: Transfer between different planning tasks. Results reflect % solved problems.

| Model | n7tree $\rightarrow$ n15star Valuepath | BW $\rightarrow$ Mystery BW | ToH $\rightarrow$ Mystery BW |
|---|---|---|---|
| GPT-4 ICL | 51 | 0.2 | 0 |
| GPT-4 CoT | 65 | 1.4 | 0 |
| **MAP** | **80** | **12.2** | **6.6** |

## 4.1 ABLATION STUDY

We also carried out an ablation study to determine the relative importance of each of MAP's major components, focusing on the 3-disk ToH problems. Figure 3 (left) shows the results. We found that the Monitor was the most important component, as ablating this module resulted in significantly fewer solved problems, due primarily to an increased tendency to propose invalid moves (31% invalid moves vs. 0% for other ablation models). This highlights the importance of having a separate, modularized monitoring process. Ablating the tree search and TaskDecomposer module also resulted in significantly fewer solved problems. The impaired performance following the ablation of the tree search indicates the benefit of considering the multi-step implications of proposed actions, rather than committing to a single action as is done in standard autoregressive methods such as chain-of-thought. The impaired performance following the ablation of the TaskDecomposer highlights the benefit of decomposing a task into subgoals, which allows MAP to factorize a complex task into a set of smaller, more manageable tasks. Overall, these results suggest that all major components played an important role in MAP's performance. Moreover, the improved performance was not due entirely to the use of tree search (which is shared with tree-of-thoughts (Yao et al., 2023)), but also depended on the incorporation of other modules such as the TaskDecomposer and especially the Monitor.

## 5 RELATED WORK

Early work in AI formalized planning as a problem of search through a combinatorial state space, typically utilizing various heuristic methods to make this search tractable (Newell & Simon, 1956;

---

[1] Results reflect % solved problems for a single run.

[2] Results reflect accuracy on a fixed random subset of 100 questions averaged over 3 runs ($\pm$ standard error).

[3] Yao et al. (2023) reported performance of 83%, but since the subset of 100 questions they used for evaluation is unknown, we ran ToT using the publicly released code on a fixed subset of 100 questions for fair comparison with MAP.

[4] Geva et al. (2021) reported human performance on a random subset of 100 questions.

Newell et al., 1959). Problems such as ToH figured prominently in this early research (Simon, 1975), as it affords the opportunity to explore ideas based on hierarchical or recursive planning (in which a larger problem is decomposed into a set of smaller problems). Our proposed architecture adopts some of the key ideas from this early work, including tree search and hierarchical planning.

A few recent studies have investigated planning and multi-step decision making in LLMs. These studies suggest that, although LLMs can perform relatively simple planning tasks (Huang et al., 2022), and can learn to make more complex plans given extensive domain-specific fine-tuning (Pallagani et al., 2022; Wu et al., 2023), they struggle on tasks that require zero-shot or few-shot generation of multi-step plans (Valmeekam et al., 2023; Momennejad et al., 2023). These results also align with studies that have found poor performance in tasks that involve other forms of extended multi-step reasoning, such as arithmetic (Dziri et al., 2023). Our approach is in large part motivated by the poor planning and reasoning performance exhibited by LLMs in these settings.

Some recent approaches have employed various forms of heuristic search to improve performance in LLMs (Lu et al., 2021; Zhang et al., 2023b), but these approaches have generally involved search at the level of individual tokens. Importantly, this is in contrast to our approach, in which search is performed at the more abstract level of task states (described in natural language). Ours is similar to other recently proposed black-box approaches in which 'thoughts' – meaningful chunks of natural language – are utilized as intermediate computations to solve more complex problems. These approaches include scratchpads (Nye et al., 2021), chain-of-thought (Wei et al., 2022b), approaches that combine LLMs with tree search (Yao et al., 2023; Zhao et al., 2024; Hao et al., 2023), reflexion (Shinn et al., 2023), agent-based and multi-agent approaches (Du et al., 2023; Zhang et al., 2023a; Wang et al., 2023c; Li et al., 2023; Wang et al., 2023a;b; Zhou et al., 2023; Prasad et al., 2023; Liu et al., 2023; Song et al., 2023), and methods for combining planning with external tools (Ruan et al., 2023; Kong et al., 2023). All of these approaches can be viewed as implementing a form of controlled, or 'system 2', processing (as contrasted with automatic, or 'system 1', processing) (Schneider & Shiffrin, 1977; Sloman, 1996; Kahneman, 2011). Our approach has a similar high-level motivation, and shares some components with other black box approaches (e.g., tree search (Yao et al., 2023)), but also introduces a number of new components (error monitoring, task decomposition, task coordination, state/action distinction), and combines these components in a novel manner (see Section A.2 for further discussion).

There have also been a number of proposals for incorporating modularity into deep learning systems, including neural module networks (Andreas et al., 2016), and recurrent independent mechanisms (Goyal et al., 2019). Ours is distinguished from these approaches by the use of black-box modules that perform specific high-level functions (many of which are inspired by formal theories of decision-making, as discussed below), rather than merely incorporating a general bias toward modularity.

Our approach is inspired by formal theories of decision-making and planning, and has a particularly close connection to reinforcement learning (RL) (Sutton & Barto, 2018). In particular, many of the modules in the MAP algorithm are closely related to aspects of traditional RL algorithms. Specifically, the Actor and Evaluator modules bear some resemblance to the actor and the critic in the popular actor-critic framework (Barto et al., 1983), and the TaskDecomposer is related to hierarchical RL (Sutton et al., 1999; Dietterich, 2000; Bacon et al., 2017), in which temporal abstractions are learned to achieve subgoals. The Predictor is also closely related to the world model that can substitute for direct interation with the environment in model-based RL (Sutton & Barto, 2018; Daw, 2012). An important difference in each of these cases is that the modules in MAP only receive a task description and a couple of examples, relying on the general-purpose knowledge of the LLM to effectively perform the task, rather than being trained through RL. This also distinguishes the approach from other recent efforts to combine LLMs and RL (Carta et al., 2023; Zhou et al., 2024; Zhai et al., 2024), which involve training with RL through direct interaction with an external environment, whereas MAP generates plans internally. Finally, there are also some modules that have no obvious analog in previous RL algorithms, but which were necessitated by weaknesses that LLMs display in the planning domain. These include the Monitor, which was necessitated by the tendency of LLMs to hallucinate or violate task constraints, and the Orchestrator, which allows MAP to autonomously determine when a goal has been achieved (without groundtruth evaluation) and thus terminate planning.

## 6 CONCLUSION AND FUTURE DIRECTIONS

In this work, we have proposed the MAP architecture, a modular agentic approach aimed at improving planning with LLMs. In experiments on four challenging domains, we found that MAP significantly improved multi-step planning and decision-making performance over other LLM methods (e.g., Chain of Thought, Multi-Agent Debate, Tree of Thought). While these results represent a significant step forward, there is still room for improvement. In particular, while improving performance significantly, in this work we only considered problems in which the environment is fully observable and deterministic. Future work should investigate how to extend the proposed approach to more complex open-ended environments, especially by incorporating memory mechanisms for storing knowledge about the environment as it is accumulated. Additionally, the model still has less than optimal performance on Tower of Hanoi, the Reward Revaluation graph traversal task, and the PlanBench benchmark (Valmeekam et al., 2023) (see Section A.6 for discussion of failure modes). This may be due in part to the inherent limitations of prompting and in-context learning as methods for the specialization of MAP's modules. A promising avenue for further improvement may be to jointly fine-tune smaller open-source LLMs to serve as modules across a range of diverse tasks, rather than relying only on black box methods (as with GPT-4). This approach would also eliminate the need for task-specific prompts, and may further improve zero-shot planning on novel tasks.

An additional limitation of the current implementation is the computational cost incurred by the model (see Section A.8). Although this aligns well with the deliberative nature of controlled (i.e., 'system 2') processes (Kahneman, 2011), it would nevertheless be desirable to find ways to reduce these costs. In Section A.8, we present results from a version of MAP that achieves significantly improved efficiency, while retaining the same level of performance, by caching and re-using the outputs of some modules. We also found that MAP is effective when used with a smaller model (Llama3-70B), though performance was not as strong as the version that used GPT-4. Further improvements may result from fine-tuning smaller models to perform the specialized roles of each module.

Finally, it is interesting to consider the close parallels between our proposed approach and the neural basis of human planning and decision-making. In the human brain, planning is generally thought to depend on the prefrontal cortex (PFC) (Owen, 1997; Momennejad et al., 2018; Momennejad, 2020; Russin et al., 2020; Brunec & Momennejad, 2022; Mattar & Lengyel, 2022), a region in the frontal lobe that is notably most developed in humans and is broadly involved in executive function, decision-making, and reasoning (Miller & Cohen, 2001). Research in cognitive neuroscience has revealed the involvement of several subregions or modules within the PFC that appear to be specialized to perform certain functions, many of which are closely aligned with some of the modules in our proposed approach. These include the Anterior Cingulate Cortex, which is known to play a role in conflict monitoring (Botvinick et al., 1999), similar to our Monitor module; the Orbitofrontal Cortex, which plays a role in state prediction and state evaluation (Wallis, 2007; Schuck et al., 2016), similar to our Predictor and Evaluator modules; and the Anterior PFC, which plays a role in task decomposition and coordination, similar to our TaskDecomposer and Orchestrator modules. Human planning then emerges through the coordinated and recurrent interactions among these specialized subregions, and, similar to our approach, the algorithms implemented via these interactions are closely related to RL (O'doherty, 2004; Daw et al., 2005; Valentin et al., 2007; Takahashi et al., 2011; Silvetti et al., 2014; Brunec & Momennejad, 2022; Wang et al., 2018; Botvinick et al., 2019). An exciting direction for future work is to consider how the present approach might further contribute to understanding the brain basis of planning and decision-making.

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

# A APPENDIX

## A.1 SUPPLEMENTARY ALGORITHMS

---

**Algorithm 3: Search loop.** Tree search with a depth of $L$ layers, with $B$ branches at each layer $l$. For each branch, a proposed action is sampled, and the $\text{Predictor}$ predicts the next state $\tilde{x}$. This process continues recursively until the terminal layer $L$, at which point the value $v_{l=L}$ of the terminal states is estimated by the $\text{Evaluator}$. The values are backpropagated to their parent states in the first layer, and the action that leads to the most valuable state is selected. In our implementation, we accelerate this process by caching the actions and predicted states from deeper search layers and then reusing them in subsequent searches. We also employ the $\text{Orchestrator}$ to prematurely terminate search if the goal state is achieved.

---

**Function** Search $(l, L, B, x, y)$ **:**

    $V_l \leftarrow \{\}$                 `// Initialize value record`

    $\tilde{X}_l \leftarrow \{\}$              `// Initialize next-state record`

    $A_l \leftarrow \text{ProposeAction}(x, y, B)$         `// Propose B actions`

    **for** $b$ in $1 \ldots B$ **do**

        $\tilde{x}_{lb} \leftarrow \text{Predictor}(x, A_{lb})$        `// Predict next state`

        $\tilde{X}_l \leftarrow \tilde{X}_l \cup \{\tilde{x}_{lb}\}$        `// Update next-state record`

        $\Omega \leftarrow \text{Orchestrator}(\tilde{x}_{lb}, y)$     `// Terminate search if goal achieved`

        **if** $l < L$ and $\Omega$ is false **then**

            $a_{l+1}, \tilde{x}_{l+1}, v_{l+1} \leftarrow \text{Search}(l+1, L, B, \tilde{x}_{lb}, y)$     `// Advance search depth`

            $V_l \leftarrow V_l \cup \{v_{l+1}\}$        `// Update value record`

        **else**

            $v_{lb} \leftarrow \text{Evaluator}(\tilde{x}_{lb}, y)$        `// Evaluate predicted state`

            $V_l \leftarrow V_l \cup \{v_{lb}\}$        `// Update value record`

        **end**

    **end**

    $v_l \leftarrow \max(V_l)$        `// Maximum value (randomly sample if equal value)`

    $a_l \leftarrow A_{l\,\text{argmax}(V_l)}$        `// Select action`

    $\tilde{x}_l \leftarrow \tilde{X}_{l\,\text{argmax}(V_l)}$        `// Predicted next-state`

**return** $a_l, \tilde{x}_l, v_l$

---

## A.2   EXTENDED RELATED WORK

In this section, we consider in more detail how MAP relates to existing black-box and agentic LLM approaches:

- Similar to MAP, both scratchpad (Nye et al., 2021) and chain-of-thought (CoT) (Wei et al., 2022b) decompose a problem into intermediate computations. However, unlike MAP, neither scratchpad nor CoT factorize these intermediate computations into specialized modules.

- Tree-of-thought (ToT) (Yao et al., 2023) introduces some degree of factorization, but the factorization is not as extensive as in MAP. The 'generator' module in ToT carries out a combination of the functions carried out by both the Actor (action proposal) and the Predictor (prediction of the states that will result from these actions) in MAP. The 'evaluator' module in ToT carries out a combination of the functions carried out by both the Monitor (error detection) and the Evaluator (prediction of state value) in MAP. ToT does not contain any component that carries out the functions of the TaskDecomposer (subgoal proposal) and the Orchestrator (autonomously determining when a goal or subgoal has been achieved). LLM-MCTS (Zhao et al., 2024) and RAP (Hao et al., 2023) also propose similar approaches involving the combination of LLMs with tree search.

- Multi-agent debate (i.e., Society of Mind) (Du et al., 2023) involves the interaction of multiple LLM instances; but, unlike MAP, these model instances are not specialized to perform specific functions. Similarly, the Large Language Model-based Actor-Critic (LLaMAC) approach (Zhang et al., 2023a) involves interaction between many LLM agents, and incorporates more specialization (there is a 'critic' module that coordinates the decision-making process across many 'actors'), though there is less module specialization than there is in MAP. Similarly, CAMEL (Li et al., 2023) involves interaction between two role-playing modules (a user and an assistant).

- Similar to MAP, reflexion (Shinn et al., 2023) involves an element of self evaluation of proposed policies, but this depends on interaction with the external environment to determine the outcome of each policy (whereas in MAP, this self evaluation process is entirely internal to the agent). The dependence on interaction with the external environment makes this approach unsuitable for the planning domain (planning is, by definition, performed internally).

- Describe-Explain-Plan-Select (Wang et al., 2023c) involves the coordination of multiple modules, but the approach is specific to settings involving an agent that is spatially embedded in a 2D environment. For instance, the method utilizes the spatial proximity of objects to the agent for prioritization of subgoals. This approach cannot be directly applied to the tasks that we consider in the present work.

- JARVIS-1 (Wang et al., 2023b), Voyager (Wang et al., 2023a), BOLAA (Liu et al., 2023), and LLM-Planner (Song et al., 2023) all involve agent-based approaches featuring multiple LLM modules, but these approaches require interaction with an external environment to iteratively solve a problem, rather than developing a plan internally as in our approach.

- Adapt (Prasad et al., 2023) is another modular agent-based approach that, similar to our approach, involves task decomposition, and LATS (Zhou et al., 2023) is another agent-based approach that employs tree search. Both of these approaches require interaction with an external environment to develop plans, whereas plans are developed internally in our approach. Furthermore, Adapt doesn't use any tree search, and LATS doesn't decompose a task into subgoals, both of which are present in MAP.

- Some recent work has aimed to combine external tool use with LLM planning agents, including Task Planning and Tool Usage (TPTU) (Ruan et al., 2023; Kong et al., 2023). Because this work evaluates on tasks involving external tool use, it is not directly comparable with our approach, but we plan to extend our approach to incorporate tool use in future. We expect that MAP's modular approach will afford improved performance in this domain, relative to the more minimal planning approaches that have previously been employed (i.e., involving planning in a single LLM agent).

### A.3 Experiment Details

We implemented each of the modules using a separate GPT-4 (32K context, '2023-03-15-preview' model index for ToH and cogeval tasks, and 128K context, '0125-preview' model index for strategyQA and planbench tasks from Microsoft Azure openAI service) instance through a combination of prompting and few-shot in-context examples. We set Top-p to 0 and temperature to 0, except for the $\text{Actor}$ (as detailed in section A.7.2). The $\text{Search}$ loop explored $B = 2$ branches recursively for a depth $L = 2$.

For ToH, we used two randomly selected in-context examples of three-disk problems, and a description of the problem in the prompts for all the modules. For the graph traversal tasks, we used two in-context examples for all modules, except for the $\text{Actor}$ and $\text{Evaluator}$ in the Steppath task, where we used three in-context examples, one each for 2-, 3-, and 4-step paths. For strategyQA, we didn't use any in-context examples. For the logistics task from Planbench, we used two incontext examples for all modules except for $\text{Actor}$ which used three in-context examples. For the mystery blocksworld (deceptive) task from Planbench, we used two incontext examples for all modules except for $\text{Actor}$ and $\text{Predictor}$ which used three in-context examples. For both the tasks from Planbench, we extracted the goal from the initial state conditions, and the state and the goal was separately fed as input to the modules as required. The prompt also described the specific task that was to be performed by each module (e.g., monitoring, task decomposition). For more details about the prompts and specific procedures used for each module, see Section A.7.

For three-disk problems, we allowed a maximum of $T = 10$ actions per problem, and evaluated on 24 out of 26 possible problems (leaving out the two problems that were used as in-context examples for the $\text{Actor}$). We also evaluated on four-disk problems, for which we allowed a maximum of $T = 20$ actions per problem. The same three-disk problems were used as in-context examples, meaning that the four-disk problems tested for out-of-distribution (OOD) generalization. For the graph traversal tasks, we allowed a maximum of $T = 6$ actions per problem. For strategyQA we allowed $T = 1$ action per problem. For the Planbench tasks we allowed a maximum of $T = 4 +$ number of actions in the ground truth plan.

We didn't use a separate $\text{Predictor}$ for the graph traversal tasks, since the action proposed by the $\text{Actor}$ gives the next state. We also did not include the $\text{TaskDecomposer}$ for these tasks, and did not use the $\text{Search}$ loop for the Steppath task, as the model's performance was already at ceiling without the use of these components. For strategyQA we didn't use the $\text{Evaluator}$ or the $\text{Orchestrator}$. For the Planbench tasks we didn't use tree search, and for mystery blocksworld task we didn't use the $\text{TaskDecomposer}$.

## A.4 ADDITIONAL DESCRIPTION OF PLANNING TASKS

**Graph Traversal.** We focused on a particular type of graph (Figure 4) with community structure (Schapiro et al., 2013) previously found to be challenging for a wide variety of LLMs. The first task, Valuepath, involves finding the shortest path from a given room to the room with the largest reward, while avoiding the room that has a smaller reward. A smaller reward and a larger reward are located at two different positions in the graph. We fixed the two reward locations, and created 13 problems based on different starting locations. The second task, Steppath, involves finding the shortest path between a pair of nodes. We evaluated problems with an optimal shortest path of 2, 3, or 4 steps. We generated 20 problems for each of these conditions by sampling different starting and target locations.

The other two tasks, Detour and Reward Revaluation, involve modifications to the Valuepath task that test for flexibility in planning. In these tasks, the problem description and in-context examples for the Valuepath task are presented, and a single Valuepath problem is solved as in the original task. The task is then modified in-context in one of two ways. In the Detour task, an edge is removed from the graph and replaced with a new edge (e.g., 'the door from room 1 to room 11 is locked and now room 13 is connected to room 11'). In the Reward Revaluation task, the value associated with the two reward locations is changed (e.g., 'the reward of the chest in room 8 has been changed to 12 and the reward of the chest in room 15 has been changed to 48'). As with the Valuepath task, the Detour and Reward Revaluation tasks each involved 13 problems based on different starting locations.

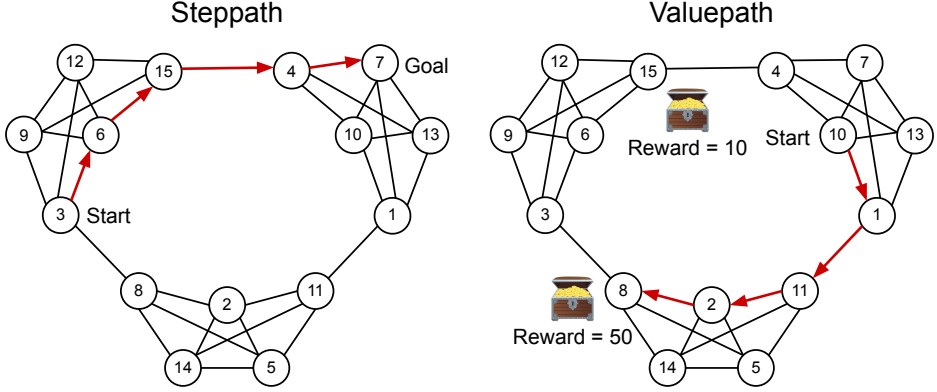

Figure 4: **Graph Traversal.** We investigated two graph traversal tasks utilizing a challenging graph with community structure. **Steppath:** Find shortest path between two nodes, e.g. node 3 and node 7. **Valuepath:** Find shortest path from starting location (e.g., node 10) to location with maximum reward (node 8 in depicted example).

**Tower of Hanoi.** In the original version of the Tower of Hanoi (ToH) task, there are three pegs and a set of disks of different sizes. The disks are stacked in order of decreasing size on the leftmost peg. The goal is to move all disks to the rightmost peg, such that the disks are stacked in order of decreasing size. There are a couple of rules that determine which moves are considered valid. First, a disk can only be moved if it is at the top of its stack. Second, a disk can only be moved to the top of another stack if it is smaller than the disks in that stack (or if the peg is empty). More complex versions of the task can be created by using a larger number of disks.

We designed an alternative formulation of this task in which the inputs are text-based rather than visual. In this alternative formulation, three lists (A, B, and C) are used instead of the three pegs, and a set of numbers (0, 1, 2, and so on) is used instead of disks of different sizes. The goal is to move all numbers so that they are arranged in ascending order in list C. The rules are isomorphic to ToH. First, a number can only be moved if it is at the end of a list. Second, a number can only be moved to the end of a new list if it is larger than all the numbers in that list. Note that although this novel formulation is isomorphic to ToH (and equally complex), it does not share any surface features with the original ToH puzzle (disks, pegs, etc.), and thus GPT-4 cannot rely on exposure to descriptions of ToH in its training data to solve the problem. We created multiple problem instances

by varying the initial state (the initial positions of the numbers). This resulted in 26 three-disk problems and 80 four-disk problems.

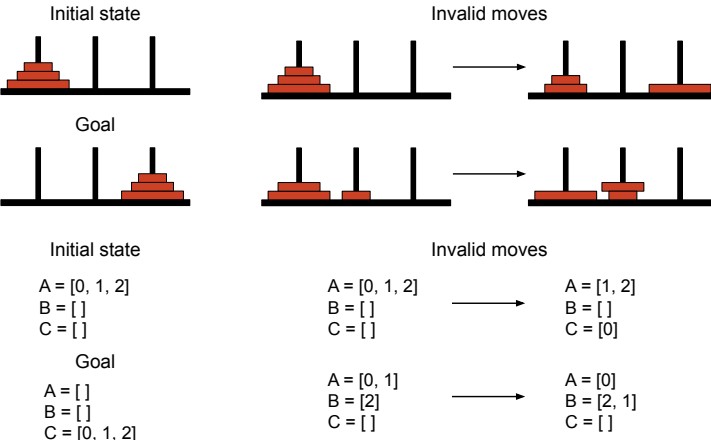

Figure 5: **Tower of Hanoi. Top:** Depiction of the Tower of Hanoi (ToH) puzzle. Disks are stacked in order of decreasing size on the leftmost peg. The goal is to move these disks so that they are stacked in order of decreasing size on the rightmost peg. Only the disk on the top of the stack may be moved, and a disk can only be placed on top of larger disks (or on an empty peg). The version shown involves three disks, but more disks can be used (making the task significantly more difficult). **Bottom:** Modified text-based version of ToH. Three lists are presented, labelled A, B and C. A set of integers is distributed amongst these lists. The goal is to move the numbers so that they are arranged in ascending order in list C. Only the number at the end of the list may be moved, and a number can only be placed in front of a smaller number. Multiple problem instances were created by varying the initial state.

## A.5   SUPPLEMENTARY RESULTS

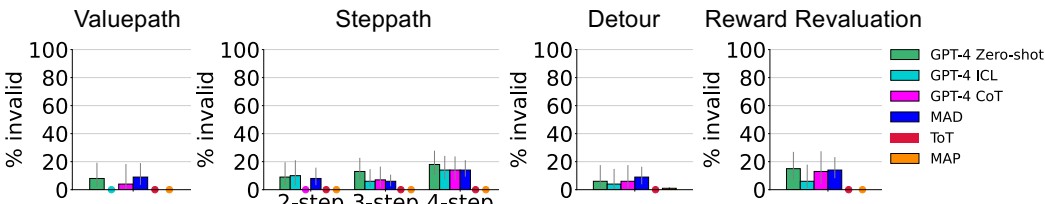

Figure 6: **Graph traversal results.** '% invalid' indicates percentage of moves that are invalid (↓ better). GPT-4 Zero-shot, ICL, CoT, and MAD baselines are deterministic, and therefore a single run was performed on all problems. Note that MAP did not employ tree search on the Steppath task, and did not employ task decomposition on any of the graph traversal tasks. Without tree search, MAP's performance is deterministic, and therefore only a single run was performed on the Steppath task, whereas we performed 5 runs with ToT. Gray error bars reflect 95% binomial confidence intervals (for models evaluated on a single run). Dots reflect values of 0%. For Valuepath, Detour, and Reward Revaluation we performed 10, 10, and 5 runs respectively with MAP and ToT, and present average performance ± the standard error of the mean (black error bars).

Table 4: Results on Valuepath task. Values within brackets indicate best performance.

| Model | % solved problems | % invalid actions | Avg plan steps | | |
|---|---|---|---|---|---|
| | | | 1-step | 2-step | 4-step |
| GPT-4 Zero-shot | 54 | 8 | 2.5 | 2.5 | 5 |
| GPT-4 ICL | 91 | **0** | 1.75 | 2.33 | **4.67** |
| GPT-4 CoT | 91 | 4 | | | |
| MAD | 73 | 9 | | | |
| ToT | 55(**100**) | **0** | | | |
| **MAP** | **100** | **0** | **1.5** | **2** | 4.75 |

Table 5: Results on Steppath task. Values within brackets indicate best performance.

| Model | % solved problems | | | % invalid actions | | | Avg plan steps | | |
|---|---|---|---|---|---|---|---|---|---|
| | 2-step | 3 step | 4-step | 2-step | 3-step | 4-step | 2-step | 3-step | 4-step |
| GPT-4 Zero-shot | 75 | 40 | 20 | 9 | 13 | 18 | **2.07** | 4 | 5.25 |
| GPT-4 ICL | 74 | 74 | 42 | 10 | 6 | 14 | 2.14 | 3.78 | **4.38** |
| GPT-4 CoT | 95 | 79 | 47 | 0 | 7 | 14 | | | |
| MAD | 82 | 79 | 39 | 8 | 6 | 14 | | | |
| ToT | 67(**100**) | 67(89) | 50(63) | **0** | **0** | **0** | | | |
| **MAP** | **100** | **100** | **95** | **0** | **0** | **0** | 2.1 | **3.42** | 4.5 |

Table 6: Results on Detour task. Values within brackets indicate best performance.

| Model | % solved problems | % invalid actions |
|---|---|---|
| GPT-4 Zero-shot | 54 | 6 |
| GPT-4 ICL | 38 | 4 |
| GPT-4 CoT | 69 | 6 |
| MAD | 38 | 9 |
| ToT | 33 (**100**) | **0** |
| **MAP** | **85** (**100**) | 1 |

Table 7: Results on Reward Revaluation task. Values within brackets indicate best performance.

| Model | % solved problems | % invalid actions |
|---|---|---|
| GPT-4 Zero-shot | 31 | 15 |
| GPT-4 ICL | 54 | 6 |
| GPT-4 CoT | 54 | 13 |
| MAD | 46 | 14 |
| ToT | 36 (**77**) | **0** |
| **MAP** | **48** (69) | **0** |

Table 8: Results on ToH. Note that we also include results here for the GPT-4 ICL baseline when prompted with 5 ICL examples (as opposed to 2 examples in the standard version of the baseline). Surprisingly, more ICL examples hurts performance on this task, perhaps due to overfitting to the specific examples (Hasanbeig et al., 2023). Values within brackets indicate best performance.

| Model | % solved problems | | % invalid actions | |
|---|---|---|---|---|
| | 3-disk | 4-disk (OOD) | 3-disk | 4-disk (OOD) |
| GPT-4 Zero-shot | 11 | 2 | 30 | 50 |
| GPT-4 ICL | 46 | 1 | 12 | 41 |
| GPT-4 ICL (5 examples) | 38 | 1 | 19 | 41 |
| GPT-4 CoT | 42 | 5 | 22 | 39 |
| MAD | 25 | 1 | 24 | 43 |
| ToT-MAP | 6 (25) | 1(4) | 4 | 5 |
| ToT | 6 (25) | 0 (0) | 14 | 11 |
| **MAP** | **74** (**100**) | **24** | **0** | **0** |

Table 9: Ablation study on ToH with 3 disks. Values within brackets indicate best performance.

| Model | % solved problems | % invalid actions |
|---|---|---|
| **MAP** | **74** (**100**) | **0** |
| w/o Task Decomposer | 50 (67) | **0** |
| w/o Tree Search | 32 (42) | **0** |
| w/o Monitor | 27 (33) | 31 |

Table 10: Results on ToH with 3 disks with a smaller LLM (Llama3-70B). MAP with Llama3-70B even outperforms the best GPT-4 baseline (GPT-4 ICL).

| Model | % solved problems | % invalid actions |
|---|---|---|
| Llama3-70B Zero-shot | 19.2 | 33.8 |
| Llama3-70B ICL | 12.5 | 41.4 |
| Llama3-70B CoT | 29.2 | 33.3 |
| GPT-4 ICL | 46 | 12 |
| **Llama3-70B MAP** | **50** | **2** |

Table 11: Results on Mystery Blocksworld domain of PlanBench using a zero-shot version of MAP (no in-context examples were provided for any module).

| Model | % solved problems |
|---|---|
| GPT-4 Zero-shot | 0.2 |
| GPT-4 ICL | 7.8 |
| **MAP Zero-shot** | **8.2** |

## A.6 ANALYSIS OF FAILURE MODES

To better understand the failure modes displayed by MAP, we analyzed the log files for the Tower of Hanoi (ToH) task (3-disk problems, run accuracy=75%, failed to solve 6/24 problems). We identified the following three general failure modes:

1. Incorrect decomposition: failure to identify subgoals that lie along the optimal path.
2. No progress: taking actions that either move away from or do not make progress toward a subgoal.
3. Falling into loops: visiting a state more than once.

These failure modes were not mutually exclusive. Table 15 shows the number of problems (out of 6 total failures) that involved these failure modes. We then identified which modules were responsible

Table 12: Results on ToH 3 disks with baselines also provided with the goal recursion strategy (provided to the Decomposer of MAP).

| Model | % solved problems |
|---|---|
| GPT-4 Zero-shot | 11 |
| GPT-4 Zero-shot w/ goal recursion | 23 |
| GPT-4 ICL | 46 |
| GPT-4 ICL w/ goal recursion | 46 |
| GPT-4 CoT | 42 |
| GPT-4 CoT w/ goal recursion | 50 |
| **MAP w/ goal recursion** | **74** |

Table 13: Results on Planbench with ToT on a subset of problems. Values within brackets indicate best performance.

| Model | Logistics (30 problems) | Mystery BW (100 problems) |
|---|---|---|
| ToT | 10.4 (16.7) | 0.6 (3) |
| **MAP** | **53.3** | **35** |

Table 14: Results on ToH with 3 disks varying the tree search hyperparameters of MAP. Mean and standard error are reported across 10 runs for computational cost and performance.

| Model | Num. calls | Num. input tokens | Num. output tokens | % solved |
|---|---|---|---|---|
| MAP ($B = 2, L = 1$) | $32.04 \pm 1.5$ | $32{,}630.62 \pm 1{,}318.4$ | $2{,}197.37 \pm 146.7$ | $55 \pm 6$ |
| **MAP ($B = 2, L = 2$)** | **$43.46 \pm 2.4$** | **$49{,}537.37 \pm 2{,}563.7$** | **$2{,}727.55 \pm 214.2$** | **$72 \pm 2$** |
| MAP ($B = 2, L = 3$) | $65.65 \pm 3.6$ | $90{,}406.30 \pm 4{,}784.7$ | $3{,}845.88 \pm 294.8$ | $69 \pm 2$ |

for these failures. The 'incorrect decomposition' failure mode was due to mistakes by the Decomposer (2/6 failures). The 'no progress' and 'falling into loops' failure modes were due to mistakes made both by the Actor and the Evaluator. Specifically, the Actor sometimes failed to propose at least one action that made progress toward the goal, and the Evaluator sometimes failed to select the action that made progress toward the goal. The Actor and Evaluator made these mistakes at least once for each of the 6 failures. Overall, we found that failures stemmed from errors made by the Decomposer, Actor, and Evaluator modules, whereas the other modules (Monitor, Predictor, and Orchestrator) performed perfectly.

Table 15: Results for number of failures (out of 6 total failures for a given run) of MAP on ToH with 3 disks for each of the three failure modes.

| Incorrect decomposition | No progress | Falling into loops |
|---|---|---|
| 2/6 | 6/6 | 5/6 |

### A.7 PROMPTS AND IN-CONTEXT EXAMPLES

#### A.7.1 TASK DECOMPOSER

For ToH, the TaskDecomposer generated a single subgoal per problem. The in-context examples included chain-of-thought reasoning (Wei et al., 2022b) based on the goal recursion strategy (Simon, 1975), which is sometimes provided to human participants in psychological studies of problem solving (Carpenter et al., 1990). The specific prompt and in-context examples are shown below:

```
Consider the following puzzle problem:

Problem description:
- There are three lists labeled A, B, and C.
- There is a set of numbers distributed among those three lists.
- You can only move numbers from the rightmost end of one list to
the rightmost end of another list.
Rule #1:  You can only move a number if it is at the rightmost end
of its current list.
Rule #2:  You can only move a number to the rightmost end of a
list if it is larger than the other numbers in that list.
A move is valid if it satisfies both Rule #1 and Rule #2.
A move is invalid if it violates either Rule #1 or Rule #2.

Goal:  The goal is to generate a single subgoal from the current
configuration, that helps in reaching the goal configuration using
minimum number of moves.

To generate subgoal use the goal recursion strategy.  First if the
smallest number isn't at the correct position in list C, then set
the subgoal of moving the smallest number to its correct position
in list C.But before that, the numbers larger than the smallest
number and present in the same list as the smallest number must
be moved to a list other than list C. This subgoal is recursive
because in order to move the next smallest number to the list
other than list C, the numbers larger than the next smallest
number and present in the same list as the next smallest number
must be moved to a list different from the previous other list and
so on.

Note in the subgoal configuration all numbers should always be in
ascending order in all the three lists.

Here are two examples:

Example 1:

This is the current configuration:
A = [0,1]
B = [2]
C = []

This is the goal configuration:
A = []
B = []
C = [0, 1, 2]

Answer:
I need to move 0 from list A to list C.
Step 1.  Find the numbers to the right of 0 in list A. There is 1
to the right of 0.
Step 2.  Find the numbers larger than 0 in list C. There are none.
I will move the numbers found in Step 1 and Step 2 to list B.
```

```
Hence I will move 1 from list A to list B. Also numbers should
be in ascending order in list B.
Subgoal:
A = [0]
B = [1, 2]
C = []

Example 2:

This is the current configuration:
A = [1]
B = [0]
C = [2]

This is the goal configuration:
A = []
B = []
C = [0, 1, 2]

Answer:
I need to move 0 from list B to list C.
Step 1.  Find the numbers to the right of 0 in list B. There are
none.
Step 2.  Find the numbers larger than 0 in list C. There is 2
which is larger than 0.
I will move the numbers found in Step 1 and Step 2 to list A.
Hence, I will move 2 from list C to list A. Also numbers should
be in ascending order in list A.
Subgoal:
A = [1, 2]
B = [0]
C = []

Here is the task:

This is the current configuration:
A = [0, 1, 2]
B = []
C = []

This is the goal configuration:
A = []
B = []
C = [0, 1, 2]

Answer:
```

A.7.2 ACTOR

The Actor was prompted to propose $B = 2$ distinct actions. In some instances, the Actor failed to propose two distinct actions. In those cases, we iteratively scaled the temperature by a factor of 0.1. This was done for a maximum of 10 attempts or until two distinct actions were produced. If the Actor was not able to propose two distinct actions even after 10 attempts, we then used only a single action. Please note that although the in-context examples involve both actions and resulting states, the Actor is only responsible for proposing actions. The specific prompt and in-context examples for the ToH task are shown below:

```
Consider the following puzzle problem:

Problem description:
- There are three lists labeled A, B, and C.
```

- There is a set of numbers distributed among those three lists.
- You can only move numbers from the rightmost end of one list to
the rightmost end of another list.
Rule #1:  You can only move a number if it is at the rightmost end
of its current list.
Rule #2:  You can only move a number to the rightmost end of a
list if it is larger than the other numbers in that list.
A move is valid if it satisfies both Rule #1 and Rule #2.
A move is invalid if it violates either Rule #1 or Rule #2.

Goal:  The goal is to end up in the goal configuration using
minimum number of moves.

Here are two examples:

Example 1:

This is the starting configuration:
A = [0, 1]
B = [2]
C = []
This is the goal configuration:
A = []
B = []
C = [0, 1, 2]

Here is the sequence of minimum number of moves to reach the goal
configuration from the starting configuration:

Move 2 from B to C.
A = [0, 1]
B = []
C = [2]

Move 1 from A to B.
A = [0]
B = [1]
C = [2]

Move 2 from C to B.
A = [0]
B = [1, 2]
C = []

Move 0 from A to C.
A = []
B = [1, 2]
C = [0]

Move 2 from B to A.
A = [2]
B = [1]
C = [0]

Move 1 from B to C.
A = [2]
B = []
C = [0, 1]

Move 2 from A to C.
A = []
B = []
C = [0, 1, 2]

```
Example 2:

This is the starting configuration:
A = [1]
B = [0]
C = [2]
This is the goal configuration:
A = []
B = []
C = [0, 1, 2]

Here is the sequence of minimum number of moves to reach the goal
configuration from the starting configuration:

Move 2 from C to A.
A = [1, 2]
B = [0]
C = []

Move 0 from B to C.
A = [1, 2]
B = []
C = [0]

Move 2 from A to B.
A = [1]
B = [2]
C = [0]

Move 1 from A to C.
A = []
B = [2]
C = [0, 1]

Move 2 from B to C.
A = []
B = []
C = [0, 1, 2]

Here is the task:

This is the starting configuration:
A = [0, 1, 2]
B = []
C = []
This is the goal configuration:
A = [0]
B = [1, 2]
C = []
Give me only two different valid next moves possible from the
starting configuration that would help in reaching the goal
configuration using as few moves as possible.
Your answer should be in the format as below:
1.  Move <N> from <src> to <trg>.
```

### A.7.3 MONITOR

The Monitor was prompted with chain-of-thought reasoning in which each of the rules of the task were checked before determining action validity. We stored the actions deemed valid by the Monitor in a separate buffer, and we terminated the action proposal loop (Algorithm 1) when there were two distinct actions in this buffer, or exceeded a maximum of 10 interactions with the Monitor. After

termination of the action proposal loop, if the buffer didn't contain two distinct actions, we used the only action in the buffer. If the buffer was empty, we used the action(s) proposed by the Actor at the last attempt. The following text was used as a prompt and in-context examples in the ToH task:

```
Consider the following puzzle problem:

Problem description:
- There are three lists labeled A, B, and C.
- There is a set of numbers distributed among those three lists.
- You can only move numbers from the rightmost end of one list to
the rightmost end of another list.
Rule #1:  You can only move a number if it is at the rightmost end
of its current list.
Rule #2:  You can only move a number to the rightmost end of a
list if it is larger than the other numbers in that list.
A move is valid if it satisfies both Rule #1 and Rule #2.
A move is invalid if it violates either Rule #1 or Rule #2.

Goal:  The goal is to check if the proposed move satisfies or
violates Rule #1 and Rule #2 and based on that if it is a valid
or invalid move.

Here are two examples:

Example 1:

This is the initial configuration:
A = []
B = [1]
C = [0, 2]

Proposed move:
Move 0 from C to B.

Answer:
First check whether the move satisfies or violates Rule #1.  Index
of 0 in list C is 0.  Length of list C is 2.  The difference in
length of list C and index of 0 in list C is 2, which is not equal
to 1.  Hence 0 is not at the rightmost end of list C, and the move
violates Rule #1.
Next check whether the move satisfies or violates Rule #2.  For
that compute the maximum of list B, to which 0 is moved.  Maximum
of list B is 1.  0 is not larger than 1.  Hence the move violates
Rule #2.
Since the Move 0 from list C to list B violates both Rule #1 and
Rule #2, it is invalid.

Example 2:

This is the initial configuration:
A = []
B = [1]
C = [0, 2]

Proposed move:
Move 2 from C to B.

Answer:
First check whether the move satisfies or violates Rule #1.  Index
of 2 in list C is 1.  Length of list C is 2.  The difference in
length of list C and index of 2 in list C is 1.  Hence 2 is at the
rightmost end of list C, and the move satisfies Rule #1.
```

```
Next check whether the move satisfies or violates Rule #2.  For
that compute the maximum of list B, to which 2 is moved.  Maximum
of list B is 1.  2 is larger than 1.  Hence the move satisfies
Rule #2.
Since the Move 2 from list C to list B satisfies both Rule #1 and
Rule #2, it is valid.

Here is the task:

This is the initial configuration:
A = []
B = [0, 1]
C = [2]

Proposed move:
Move 1 from B to A.

Answer:
```

### A.7.4  PREDICTOR

The Predictor was prompted to predict the next state, given the current state and the proposed action. The following text was used as a prompt and in-context examples in the ToH task:

```
Consider the following puzzle problem:

Problem description:
- There are three lists labeled A, B, and C.
- There is a set of numbers distributed among those three lists.
- You can only move numbers from the rightmost end of one list to
the rightmost end of another list.
Rule #1:  You can only move a number if it is at the rightmost end
of its current list.
Rule #2:  You can only move a number to the rightmost end of a
list if it is larger than the other numbers in that list.

Goal:  The goal is to predict the configuration of the
three lists, if the proposed move is applied to the current
configuration.

Here are two examples:

Example 1:

This is the current configuration:
A = []
B = [1]
C = [0, 2]

Proposed move:
Move 2 from list C to list B.

Answer:
A = []
B = [1, 2]
C = [0]

Example 2:

This is the current configuration:
A = []
```

```
B = [1]
C = [0, 2]

Proposed move:
Move 1 from list B to list A.

Answer:
A = [1]
B = []
C = [0, 2]

Here is the task:

This is the current configuration:
A = []
B = [0, 1]
C = [2]

Proposed move:
Move 1 from list B to list A.

Answer:
```

### A.7.5 EVALUATOR

For the Evaluator, in the ToH task, GPT-4 was prompted to generate a heuristic function that could be used to estimate the distance between the current state and the goal state. The description of this heuristic function was then included in the Evaluator's prompt. In the graph traversal tasks, we used only in-context examples, with no heuristic function prompt.

The following text shows prompt and in-context examples used in the ToH task. The first User message prompts GPT-4 to generate a heuristic function. After GPT-4's response, another User message prompts the Evaluator to use this function:

USER: ```
Consider the following puzzle problem:

Problem description:
- There are three lists labeled A, B, and C.
- There is a set of numbers distributed among those three lists.
- You can only move numbers from the rightmost end of one list to
the rightmost end of another list.
Rule #1:  You can only move a number if it is at the rightmost end
of its current list.
Rule #2:  You can only move a number to the rightmost end of a
list if it is larger than the other numbers in that list.
A move is valid if it satisfies both Rule #1 and Rule #2.
A move is invalid if it violates either Rule #1 or Rule #2.

Goal:  The goal is to predict the minimum number of valid
moves required to reach the goal configuration from the current
configuration.

Here are two examples:

Example 1:

This is the current configuration:
A = [0, 1, 2]
B = []
C = []
```

This is the goal configuration:
A = []
B = []
C = [0, 1, 2]

Answer:
The minimum number of valid moves required to reach the goal
configuration from the current configuration is 7.

Example 2:

This is the current configuration:
A = [1, 2]
B = [0]
C = []

This is the goal configuration:
A = []
B = []
C = [0, 1, 2]

Answer:
The minimum number of valid moves required to reach the goal
configuration from the current configuration is 4.

What heuristic function can be used to estimate the minimum number
of valid moves required to reach the goal configuration from a
given current configuration?

GPT-4:    A suitable heuristic function for this problem is the "sum
of the distances" heuristic.  This heuristic estimates the minimum
number of valid moves required to reach the goal configuration by
calculating the sum of the distances each number needs to travel
to reach its final position in the goal configuration.

Here's how the heuristic function works:

1.  For each number in the current configuration, determine its
current position (list and index) and its goal position (list and
index) in the goal configuration.
2.  Calculate the distance between the current position and the
goal position for each number.  The distance can be calculated
as the absolute difference between the indices of the current and
goal positions, plus a penalty if the number needs to move to a
different list.
3.  Sum the distances calculated in step 2 for all numbers.

The heuristic function will return the sum of the distances, which
is an estimate of the minimum number of valid moves required to
reach the goal configuration from the current configuration.

This heuristic is admissible because it never overestimates
the cost of reaching the goal configuration.  It considers the
minimum number of moves required for each number to reach its goal
position, without taking into account the constraints imposed by
the rules of the puzzle.  Therefore, the actual number of moves
required to reach the goal configuration will always be greater
than or equal to the heuristic value.

USER: This is the current configuration:
```
A = [0]
B = []
C = [1, 2]
```
This is the goal configuration:
```
A = [0]
B = [1, 2]
C = []
```

Use the heuristic function to predict the minimum number of valid moves required to reach the goal configuration from the current configuration.

Please provide your answer according to the heuristic function in the format as below:
The minimum number of valid moves required to reach the goal configuration from the current configuration is <N>.

### A.7.6 ORCHESTRATOR

The Orchestrator was prompted to determine whether the current subgoal was achieved by the current state. The following text was used as a prompt and in-context examples in the ToH task:

Consider the following puzzle problem:

Problem description:
- There are three lists labeled A, B, and C.
- There is a set of numbers distributed among those three lists.
- You can only move numbers from the rightmost end of one list to the rightmost end of another list.
Rule #1: You can only move a number if it is at the rightmost end of its current list.
Rule #2: You can only move a number to the rightmost end of a list if it is larger than the other numbers in that list.

Goal: The goal is to predict whether the current configuration matches the goal configuration or not.

Here are two examples:

Example 1:

This is the current configuration:
```
A = []
B = []
C = [0, 1, 2]
```

This is the goal configuration:
```
A = []
B = []
C = [0, 1, 2]
```

Answer: The current configuration matches the goal configuration. Hence yes.

Example 2:

This is the current configuration:
```
A = [0, 1]
B = [2]
C = []
```

This is the goal configuration:
```
A = []
```

```
B = []
C = [0, 1, 2]

Answer:  The current configuration doesn't match the goal
configuration.  Hence no.

Here is the task:

This is the current configuration:
A = []
B = [0, 1, 2]
C = []

This is the goal configuration:
A = []
B = []
C = [0, 1, 2]

Answer:
```

### A.7.7 ZERO-SHOT PROMPT

An example prompt for the GPT-4 zero-shot baseline is shown below:

```
Consider the following puzzle problem:

Problem description:
- There are three lists labeled A, B, and C.
- There is a set of numbers distributed among those three lists.
- You can only move numbers from the rightmost end of one list to
the rightmost end of another list.
Rule #1:  You can only move a number if it is at the rightmost end
of its current list.
Rule #2:  You can only move a number to the rightmost end of a
list if it is larger than the other numbers in that list.
A move is valid if it satisfies both Rule #1 and Rule #2.
A move is invalid if it violates either Rule #1 or Rule #2.

Goal:  The goal is to end up in the configuration where all
numbers are in list C, in ascending order using minimum number
of moves.

This is the starting configuration:
A = [0, 1, 2]
B = []
C = []
This is the goal configuration:
A = []
B = []
C = [0,1,2]

Give me the sequence of moves to solve the puzzle from the
starting configuration, updating the lists after each move.
Please try to use as few moves as possible, and make sure to
follow the rules listed above.  Please limit your answer to a
maximum of 10 steps.

Please format your answer as below:
Step 1.  Move <N> from <src> to <tgt>.
A = []
B = []
C = []
```

A.7.8  ICL PROMPT

An example prompt for the GPT-4 ICL baseline is shown below:

```
Consider the following puzzle problem:

Problem description:
- There are three lists labeled A, B, and C.
- There is a set of numbers distributed among those three lists.
- You can only move numbers from the rightmost end of one list to
the rightmost end of another list.
Rule #1:  You can only move a number if it is at the rightmost end
of its current list.
Rule #2:  You can only move a number to the rightmost end of a
list if it is larger than the other numbers in that list.
A move is valid if it satisfies both Rule #1 and Rule #2.
A move is invalid if it violates either Rule #1 or Rule #2.

Goal:  The goal is to end up in the configuration where all
numbers are in list C, in ascending order using minimum number
of moves.

Here are two examples:

Example 1:

This is the starting configuration:
A = [0, 1]
B = [2]
C = []
This is the goal configuration:
A = []
B = []
C = [0, 1, 2]

Here is the sequence of minimum number of moves to reach the goal
configuration from the starting configuration:

Move 2 from B to C.
A = [0, 1]
B = []
C = [2]

Move 1 from A to B.
A = [0]
B = [1]
C = [2]

Move 2 from C to B.
A = [0]
B = [1, 2]
C = []

Move 0 from A to C.
A = []
B = [1, 2]
C = [0]

Move 2 from B to A.
A = [2]
B = [1]
C = [0]

Move 1 from B to C.
A = [2]
```

```
B = []
C = [0, 1]

Move 2 from A to C.
A = []
B = []
C = [0, 1, 2]

Example 2:

This is the starting configuration:
A = [1]
B = [0]
C = [2]
This is the goal configuration:
A = []
B = []
C = [0, 1, 2]

Here is the sequence of minimum number of moves to reach the goal
configuration from the starting configuration:

Move 2 from C to A.
A = [1, 2]
B = [0]
C = []

Move 0 from B to C.
A = [1, 2]
B = []
C = [0]

Move 2 from A to B.
A = [1]
B = [2]
C = [0]

Move 1 from A to C.
A = []
B = [2]
C = [0, 1]

Move 2 from B to C.
A = []
B = []
C = [0, 1, 2]

Here is the task:

This is the starting configuration:
A = [0, 1, 2]
B = []
C = []
This is the goal configuration:
A = []
B = []
C = [0,1,2]

Give me the sequence of moves to solve the puzzle from the
starting configuration, updating the lists after each move.
Please try to use as few moves as possible, and make sure to
```

```
follow the rules listed above.  Please limit your answer to a
maximum of 10 steps.

Please format your answer as below:
Step 1.  Move <N> from <src> to <tgt>.
A = []
B = []
C = []
```

### A.7.9   CoT ICL PROMPT

An example prompt for the GPT-4 CoT ICL baseline is shown below:

```
Consider the following puzzle problem:

Problem description:
- There are three lists labeled A, B, and C.
- There is a set of numbers distributed among those three lists.
- You can only move numbers from the rightmost end of one list to
the rightmost end of another list.
Rule #1:  You can only move a number if it is at the rightmost end
of its current list.
Rule #2:  You can only move a number to the rightmost end of a
list if it is larger than the other numbers in that list.
A move is valid if it satisfies both Rule #1 and Rule #2.
A move is invalid if it violates either Rule #1 or Rule #2.

Goal:  The goal is to end up in the configuration where all
numbers are in list C, in ascending order using minimum number
of moves.

Here are two examples:

Example 1:

This is the starting configuration:
A = [0, 1]
B = [2]
C = []
This is the goal configuration:
A = []
B = []
C = [0, 1, 2]

Here is the sequence of minimum number of moves along with
reasoning for each move to reach the goal configuration from the
starting configuration:

I need to move 0 from A to C. But before that I need to move the
number present to the right of 0, which is 1 to B. There is a
number larger than 1 already present in list B. Hence I first need
to move 2 from B to C.
Move 2 from B to C.
A = [0, 1]
B = []
C = [2]

I need to move 0 from A to C. But before that I need to move the
number present to the right of 0, which is 1 to B.
Move 1 from A to B.
A = [0]
B = [1]
C = [2]
```

```
I need to move 0 from A to C. There is a number larger than 0
already present in list C. Hence I first need to move 2 from C
to B.
Move 2 from C to B.
A = [0]
B = [1, 2]
C = []

There is no number to the right of 0 in A, and there is no number
larger than 0 in C. Hence, I can move 0 from A to C.
Move 0 from A to C.
A = []
B = [1, 2]
C = [0]

I need to move 1 from B to C. But before that I need to move the
number present to the right of 1, which is 2 to A.
Move 2 from B to A.
A = [2]
B = [1]
C = [0]

There is no number to the right of 1 in B, and there is no number
larger than 1 in C. Hence, I can move 1 from B to C.
Move 1 from B to C.
A = [2]
B = []
C = [0, 1]

There is no number to the right of 2 in A, and there is no number
larger than 2 in C. Hence, I can move 2 from A to C.
Move 2 from A to C.
A = []
B = []
C = [0, 1, 2]

Example 2:

This is the starting configuration:
A = [1]
B = [0]
C = [2]
This is the goal configuration:
A = []
B = []
C = [0, 1, 2]

Here is the sequence of minimum number of moves along with
reasoning for each move to reach the goal configuration from the
starting configuration:

I need to move 0 from B to C. There is a number larger than 0
already present in list C. Hence I first need to move 2 from C
to A.
Move 2 from C to A.
A = [1, 2]
B = [0]
C = []

There is no number to the right of 0 in B, and there is no number
larger than 0 in C. Hence, I can move 0 from B to C.
Move 0 from B to C.
A = [1, 2]
```

```
B = []
C = [0]

I need to move 1 from A to C. But before that I need to move the
number present to the right of 1, which is 2 to B.
Move 2 from A to B.
A = [1]
B = [2]
C = [0]

There is no number to the right of 1 in A, and there is no number
larger than 1 in C. Hence, I can move 1 from A to C.
Move 1 from A to C.
A = []
B = [2]
C = [0, 1]

There is no number to the right of 2 in B, and there is no number
larger than 2 in C. Hence, I can move 2 from B to C.
Move 2 from B to C.
A = []
B = []
C = [0, 1, 2]

Here is the task:

This is the starting configuration:
A = [0, 1, 2]
B = []
C = []
This is the goal configuration:
A = []
B = []
C = [0,1,2]

Give me the sequence of moves to solve the puzzle from the
starting configuration, updating the lists after each move.
Please try to use as few moves as possible, and make sure to
follow the rules listed above.  Please limit your answer to a
maximum of 10 steps.

Please format your answer as below:
Step 1.  Move <N> from <src> to <tgt>.
A = []
B = []
C = []
```

A.8 Computational Cost: Thinking Fast and (very) Slow

Tables 16 and 17 show various cost metrics for both MAP vs. baseline models, and compares these cost metrics with performance. To address the significant computational cost of MAP, we also developed a more efficient version that cached and re-used results for redundant prompts. This was done for all modules except the Actor and the TaskDecomposer. This version of the model was significantly more efficient, while retaining the same level of performance.

Table 16: Average per-problem computational cost ($\pm$ the standard error of the mean) on ToH with 3 disks. 5 runs were done for MAP.

| Model | Num. calls | Num. input tokens | Num. output tokens | % solved |
|---|---|---|---|---|
| GPT-4 ICL | $1 \pm 0.0$ | $810.88 \pm 0.1$ | $190.38 \pm 15.4$ | 46 |
| GPT-4 CoT | $1 \pm 0.0$ | $1309.88 \pm 0.1$ | $422.42 \pm 22.8$ | 42 |
| ToT | $45.75 \pm 0.6$ | $26649.88 \pm 454.0$ | $7523.96 \pm 322.4$ | 6 |
| MAP | $148.6 \pm 8.2$ | $109,090.025 \pm 6,567.2$ | $14,543.57 \pm 844.6$ | $74 \pm 3$ |
| MAP (efficient) | $41.99 \pm 3.6$ | $47242.43 \pm 3848.4$ | $2766.34 \pm 324.5$ | $77 \pm 3$ |

Table 17: Average per-problem computational cost ($\pm$ the standard error of the mean) incurred by each module of MAP on ToH with 3 disks.

| Module | Num. calls | Num. input tokens | Num. output tokens |
|---|---|---|---|
| Actor | $24.68 \pm 1.5$ | $38,274.62 \pm 2,796.8$ | $758.69 \pm 54.1$ |
| Monitor | $46.26 \pm 2.8$ | $32,135.44 \pm 1,933.2$ | $7,131.85 \pm 429.6$ |
| Predictor | $29.48 \pm 1.6$ | $11,284.65 \pm 601.3$ | $477.48 \pm 25.7$ |
| Evaluator | $22.52 \pm 1.2$ | $17,054.15 \pm 917.6$ | $5,730.11 \pm 334.3$ |
| Orchestrator | $24.67 \pm 1.2$ | $9,553.3 \pm 466.7$ | $316.0 \pm 16.0$ |
| Task Decomposer | $1 \pm 0.0$ | $787.88 \pm 0.0$ | $129.44 \pm 2.0$ |

