# OpenReview forum: "Improving Planning with Large Language Models: A Modular Agentic Architecture"
_ICLR.cc/2025/Conference — ICLR 2025 Conference Withdrawn Submission_

### Official Review · Reviewer_qGFq · 2024-11-01

**Soundness:** 2
**Presentation:** 2
**Contribution:** 3
**Rating:** 5
**Confidence:** 4

**Summary:**

This paper proposes MAP, a modular planning architecture designed to address the limitations of LLM-based planning methods. Each module has a specific role and operates collectively within the system. Each module is inspired by concepts from cognitive neuroscience and reinforcement learning (RL). The authors demonstrate performance improvements over baselines (GPT-4 Zero-shot, GPT-4 ICL, GPT-4 CoT, ToT, MAD) in tasks such as graph traversal, Tower of Hanoi, PlanBench, and StrategyQA.

**Strengths:**

- The concept of using multiple LLM-based modules, each with a specific role in planning, is well-conceived. Additionally, grounding each module in neuroscience and RL theories strengthens the approach.
- The tasks used in the experiments, such as graph traversal and PlanBench, effectively isolate and test the planning capabilities of LLMs, demonstrating the system’s potential to address planning challenges.

**Weaknesses:**

- The primary limitation of MAP is its high computational cost, which does not seem fully justified by the observed performance gains. According to Section A.8, the number of input tokens used in MAP is 83 times higher than GPT-4 CoT. Even the efficient version of MAP is 36 times more costly.
- The explanation of MAP in the main text is insufficient, requiring reference to multiple sections in the appendix to fully understand certain modules. For example, understanding the Actor and Monitor modules requires consulting not only the main text but also Algorithm 2, Algorithm 3, and Sections A.7.2 and A.7.3 in the appendix. Additionally, inconsistencies remain in the descriptions. For instance, Algorithm 2 shows the Actor outputting an action sequence $A$, but in Section A.7.2, both action sequences and state information are included. It is unclear whether the Actor outputs only action sequences, both action and state, or relies on other modules for state information. Similarly, in Algorithm 2, the Monitor is described as taking the action sequence $A$ from the Actor and producing validity and feedback for the sequence. However, the in-context examples in Section A.7.3 show feedback and validity generated for each individual action, creating ambiguity about how the Monitor processes an entire action sequence to produce validity and feedback.
- Like many LLM-based planners, MAP relies heavily on selecting in-context examples. However, there is no detailed explanation for how these examples are chosen. The authors should clarify whether in-context examples are fixed or dynamically selected. If fixed, identifying effective examples for each module could be costly. If dynamically selected, an explanation is needed on how examples are constructed and selected.
- The Task Decomposer appears to lack a mechanism for recovering from errors in subgoal generation, meaning that an initial mistake in generating a subgoal could lead to failure regardless of subsequent steps. Additionally, MAP currently only generates single subgoals in experiments. Verification is needed to assess its effectiveness when multiple subgoals are required.
- The Predictor module seems likely to perform well only in deterministic environments. In complex or non-deterministic environments (e.g., Minecraft [1], ALFWorld [2], WebShop [3]), accurate prediction may become challenging, risking performance degradation.

[1] Fan, Linxi, et al. "Minedojo: Building open-ended embodied agents with internet-scale knowledge." *Advances in Neural Information Processing Systems* 35 (2022): 18343-18362.

[2] Shridhar, Mohit, et al. "ALFWorld: Aligning Text and Embodied Environments for Interactive Learning." *International Conference on Learning Representations*.

[3] Yao, Shunyu, et al. "Webshop: Towards scalable real-world web interaction with grounded language agents." *Advances in Neural Information Processing Systems* 35 (2022): 20744-20757.

**Questions:**

- Each module is inspired by neuroscience and RL. Is there a theoretical background that also covers interactions between these modules?
- How are in-context examples selected and constructed for each module? Are they fixed or dynamically selected according to task requirements?
- Is prompt design automated, or does it require manual configuration? Given MAP’s architecture, prompt design for all modules could be costly and time-consuming. The automatic insights collection approach used by ExpeL [4] could also serve as a useful reference.
- Why does the Task Decomposer currently generate only a single subgoal?
- The Actor’s in-context examples (A.7.2) lack feedback examples. Can the Actor effectively process feedback when it is provided by the Monitor?
- Is there a plan to test in more complex environments (e.g., Minecraft [1], ALFWorld [2], WebShop [3]), where prediction errors may occur more frequently?
- Even efficient version of MAP incurs significant computational costs. How do the authors plan to address this issue?

[4] Zhao, Andrew, et al. "Expel: Llm agents are experiential learners." *Proceedings of the AAAI Conference on Artificial Intelligence*. Vol. 38. No. 17. 2024.

---

> ### Author Response · Authors · 2024-11-21
> **Author response to Reviewer qGFq**
>
> We thank the reviewer for their helpful suggestions and comments. Below we provide a point-by-point reply to the concerns raised:
>
> **1. High computational cost of MAP**
>
> We share the reviewer’s concerns about computational cost, but we would like to note that the efficient version of MAP has costs that are on par with other inference-time compute methods, as illustrated by the comparison with tree-of-thought (3.16 per problem for the efficient version of MAP vs. 2.50 per problem for ToT).
>
>
> **2. The explanation of MAP in the main text is insufficient, requiring reference to multiple sections in the appendix to fully understand certain modules along with inconsistencies in the descriptions.**
>
> Thank you for pointing out these issues with the presentation. Due to space constraints, we found it difficult to fit all 3 algorithms in the main text. We have now moved Algorithm 2 into the main text, since the concerns identified here pertain specifically to this algorithm. Regarding the specific issues identified here, first we would like to note that the actor does not output a sequence of actions, but instead outputs a set of potential actions to take at the next time step. We have revised the description in Section 2.3 to clarify this. Regarding Section A.7.2, we have added a clarification that although the in-context examples involve both actions and resulting states, the Actor is only responsible for proposing actions (the Predictor is responsible for predicting the resulting states). Regarding the Monitor, we hope that the confusion is addressed by having clarified that the Actor does not propose action sequences but instead only proposes potential individual actions, which are then processed separately by the Monitor.
>
> **3. Like many LLM-based planners, MAP relies heavily on selecting in-context examples. However, there is no detailed explanation for how these examples are chosen. The authors should clarify whether in-context examples are fixed or dynamically selected. If fixed, identifying effective examples for each module could be costly. If dynamically selected, an explanation is needed on how examples are constructed and selected.**
>
> The selection of in-context examples is fixed. However, we would like to note that the specific examples were not selected using any explicit optimization procedure, and the same set of in-context examples was used for all of the baselines, so it is unlikely that performance is driven by the specific selection of in-context examples, and it is unlikely that this would explain the improvement relative to the baselines. We would also like to note that we have included results for a zero-shot version of the model (i.e. without in-context examples, in Table 11), which outperformed even a GPT-4 ICL baseline (with in-context examples). Overall, we do not think that the performance of the model is attributable to the specific selection of in-context examples.
>
> **4. The Predictor module seems likely to perform well only in deterministic environments. In complex or non-deterministic environments (e.g., Minecraft [1], ALFWorld [2], WebShop [3]), accurate prediction may become challenging, risking performance degradation.**
>
> Thank you for raising this important issue. We agree that it is highly non-trivial to learn a world model, especially in non-deterministic settings, and we do not claim to have solved this problem. However, it is important to note that even in fully observable and deterministic settings, where prediction is in principle substantially easier, the planning tasks that we consider are still extremely challenging for LLMs. and for the other agent-based baselines that we consider. To make progress on these issues, in this work we aimed to tackle the planning problem in a setting where prediction is relatively easier, but ongoing research on world models is also important and is indeed synergistic with our work. To clarify this aspect of our work, we have now added a subsection that describes our problem formulation to Section 2:
>
> 2.1 PROBLEM FORMULATION
>
> We consider planning problems in environments that are both deterministic and fully observable. The environment and transition dependencies are described to the agent, along with a starting state $x$ and a goal $y$, and the agent must generate a plan $P$ (a series of actions) without directly interacting with the environment. Despite the relatively limited nature of the environments we consider (fully observable, deterministic), planning problems of this sort are nevertheless extremely challenging for LLMs (Valmeekam et al., 2023; Momennejad et al., 2023).

---

> ### Author Response · Authors · 2024-11-21
> **Author response to Reviewer qGFq (continued)**
>
> **5. Each module is inspired by neuroscience and RL. Is there a theoretical background that also covers interactions between these modules?**
>
> To some extent, this has been addressed in RL (e.g., actor-critic, model-based RL, which we discuss in the Related Work), but the full fledged model involving interaction between all components has to our knowledge not previously been attempted, and is a major part of the motivation for this work.
>
> **6. The Actor’s in-context examples (A.7.2) lack feedback examples. Can the Actor effectively process feedback when it is provided by the Monitor?**
>
> Yes, even though the Actor's in-context examples don't involve any feedback examples, the feedback from the Monitor is included in the Actor's prompt and we find that it's able to effectively incorporate that feedback.
>
> **7. Is there a plan to test in more complex environments (e.g., Minecraft [1], ALFWorld [2], WebShop [3]), where prediction errors may occur more frequently?**
>
> Yes, extension of the idea to partially observable and non–deterministic domains is an important priority for future work. Our focus in the present work was to address the major shortcomings that arise even for planning in the deterministic and fully observable setting, but we plan to extend this approach in future work.
>
> **8. Even efficient version of MAP incurs significant computational costs. How do the authors plan to address this issue?**
>
> We share the concern about computational costs. We have addressed it to some extent by showing that MAP when used with a smaller LLM (Llama3-70B) outperformed baselines using the same smaller LLM, and also outperformed the best GPT-4 baseline on ToH 3 disk task (Table 10).  In future work, we also plan to address this by fine-tuning smaller models to implement the modules (which may also improve the performance of those modules through specialization), and by selectively performing search based on the expected value of additional computation, which has been explored in previous theories from neuroscience (e.g. Shenhav et al. 2013). This is an important priority for future work.
>
> *Shenhav, A., Botvinick, M. M., & Cohen, J. D. (2013). The expected value of control: an integrative theory of anterior cingulate cortex function. Neuron, 79(2), 217-240.*

---

> ### Comment · Reviewer_qGFq · 2024-11-26
>
> I appreciate the authors' detailed explanations and efforts to incorporate my feedback. To improve readability, it would be helpful if the revised portions of the manuscript were clearly highlighted, making it easier to identify the updates. I also have several remaining questions and concerns:
>
> 1. **Comments on Response 1 (High Computational Cost of MAP)**
>
> Could you clarify where the explanation for the computational cost of "3.16 per problem for the efficient version of MAP vs. 2.50 per problem for ToT" is provided?
>
> Based on Table 16, it is evident that MAP(efficient) and ToT have comparable computational costs for ToH with 3 disks (e.g., the number of input tokens for ToT is 26,649.88, while for MAP(efficient) it is 47,242.43). However, ToT appears to be ineffective for this task, as shown by its performance in Figure 3. By comparison, GPT-4 ICL and GPT-4 CoT deliver more competitive performance.
>
> When comparing MAP to GPT-4 models, the computational cost of MAP seems disproportionately high. For instance, in Table 8, GPT-4 ICL achieves a performance of 46%, while MAP achieves 74%. Yet, Table 16 shows that the number of input tokens is 810.88 for GPT-4 ICL versus 47,242.43 for MAP(efficient). Including a comparison of performance and computational cost across these methods could help better justify MAP's advantages. A discussion or visualization, such as a performance-to-cost ratio graph, would be particularly helpful.
>
> 2. **Comments on Response 2 (Explanation of MAP in the Main Text)**
>
> The clarification regarding the Actor generating a set of potential actions is appreciated and has resolved part of the concern. However, the issue of multiple, potentially unnecessary sampling attempts remains a limitation. It appears that a single action could result from up to 10 attempts.
>
> Furthermore, the clarification regarding the Monitor leaves room for confusion. In Algorithm 1, Monitor recieves $x, A$. However, it remains unclear how the Monitor process multiple proposed actions. For instance:
>
> - Does the Monitor validate each action individually using prompts such as "Proposed move: ..." as shown in A.7.3?
> - Or does it process multiple actions simultaneously and select valid ones in a single pass?
>
> 3. **Comments on Response 3 (Selection of In-Context Examples)**
>
> Thank you for clarifying that fixed examples were used and that the same examples were applied across all baselines. This addresses the concern to some extent. To further enhance clarity, it would be helpful if the manuscript explicitly mentioned that identical fixed examples were used across all experiments to avoid any ambiguity.
>
> Additionally, there is substantial evidence suggesting that the specific selection of in-context examples can significantly impact model performance. For instance, the dynamic example selection ablation in Zero-shot Planner [1] and the in-context example selection experiments in LoTa-Bench [2] demonstrate the importance of this aspect. Therefore, it is worth considering that simply replacing the examples used in GPT-4 ICL with more effective ones could potentially achieve performance levels comparable to MAP, but with significantly lower computational complexity. While conducting these experiments during the discussion period is understandably impractical, investigating this in future work could enhance the robustness and completeness of the paper.
>
> [1] Huang, Wenlong, et al. "Language models as zero-shot planners: Extracting actionable knowledge for embodied agents." *International Conference on Machine Learning*. PMLR, 2022. \
> [2] Choi, Jae-Woo, et al. "LoTa-Bench: Benchmarking language-oriented task planners for embodied agents." *arXiv preprint arXiv:2402.08178* (2024).
>
> 4. **Unaddressed concerns**
>
> The above questions remain unresolved, and the following concerns have not been addressed:
>
> - The Task Decomposer appears to lack a mechanism for recovering from errors in subgoal generation, meaning that an initial mistake in generating a subgoal could lead to failure regardless of subsequent steps. Additionally, MAP currently only generates single subgoals in experiments. Verification is needed to assess its effectiveness when multiple subgoals are required.
> - Is prompt design automated, or does it require manual configuration? Given MAP’s architecture, prompt design for all modules could be costly and time-consuming. The automatic insights collection approach used by ExpeL [4] could also serve as a useful reference.
> - Why does the Task Decomposer currently generate only a single subgoal?
>
> While I appreciate the authors’ efforts to address some of the concerns, critical points remain unresolved. For these reasons, my score remains unchanged.

---

### Official Review · Reviewer_ACfY · 2024-11-03

**Soundness:** 3
**Presentation:** 2
**Contribution:** 2
**Rating:** 5
**Confidence:** 3

**Summary:**

This paper proposes the Modular Agentic Planner (MAP), a modular agentic architecture that improves the planning capabilities of LLMs. The architecture includes components for conflict monitoring, state prediction, state evaluation, task decomposition, and orchestration. Each module is implemented using an LLM and performs a specific function in the planning process. The authors evaluate MAP on four tasks: graph traversal, Tower of Hanoi, PlanBench, and StrategyQA, and showed improvement over baselines like CoT, ToT, ICL and Zero-shot. An ablation study is performed to highlight TreeSearch and TaskDecomposer's contribution to the MAP's performance.

**Strengths:**

1. The experiments and ablation analysis are thoroughly conducted with statistical significance.
2. Inspirations from cognitive neuroscience are interesting.
3. Writing is clear and easy to follow, and the method section is clear.
4. Thorough ablation analysis on method's failure method
5. Solid results compared to the baselines provided (especially on challenging tasks like PlanBench).

**Weaknesses:**

1. Novelty and relation to prior work -- MAP seems to be a modular agentic framework with search, which is not the first time this high-level structure is introduced, and works with similar framework were not explicit discusses. See references below.
2. In relation to point 1, I find the baselines in the experiments all too weak. There are plenty of work on LLM agents that use explicit planning modules and/or MCTS. It would be more convincing if the MAP can still outperform those. If that's the case, an analysis on why MAP outperforms planning-based methods would be great to have as well.
**Minor stuff**: tables contains odd spacing, and a lot of white spaces.

Additional references:
https://arxiv.org/pdf/2308.05960
https://arxiv.org/pdf/2212.04088
https://arxiv.org/abs/2305.14078
https://arxiv.org/pdf/2305.14992

**Questions:**

See weaknesses

---

> ### Author Response · Authors · 2024-11-21
> **Author response to Reviewer ACfY**
>
> We thank the reviewer for their helpful suggestions and comments.  Below we provide a point-by-point reply to the concerns raised:
>
> **1. Novelty and relation to prior work -- MAP seems to be a modular agentic framework with search, which is not the first time this high-level structure is introduced, and works with similar framework were not explicit discusses. See references below.**
>
> Thank you for bringing these additional works to our attention. We agree that they are relevant and have added references and discussion to the paper. One major difference with our approach is that some of these approaches, specifically BOLAA and LLM-Planner, require interaction with the external environment to develop a plan, whereas our approach performs planning internally. LLM-MCTS and RAP are both very similar to tree-of-thought, which we evaluated as a baseline. Relative to these approaches, our approach is distinguished by the inclusion of additional modules (especially the orchestrator, and monitor, and task decomposer) which enable MAP to outperform tree-of-thought. We have included references to these works in the Related Work, and have included a more detailed discussion of the relationship with our approach in the Extended Related Work (Section A.2).
>
> **2. In relation to point 1, I find the baselines in the experiments all too weak. There are plenty of work on LLM agents that use explicit planning modules and/or MCTS. It would be more convincing if the MAP can still outperform those. If that's the case, an analysis on why MAP outperforms planning-based methods would be great to have as well.**
>
> We would like to note that one of the baselines that we evaluated, tree-of-thought (ToT), does include an explicit planning module with tree search, very similar to the RAP and LLM-MCTS approaches that were mentioned by the reviewer. We found that our approach outperformed ToT on all of the benchmarks that we evaluated. This is likely due to the presence of the other components in our approach (monitoring, task decomposition), as ablating these other components (but keeping tree search) resulted in worse performance. Overall, our ablation experiments and results for the ToT baseline indicate that tree search is not solely responsible for the performance gains demonstrated by the model.
>
> **3. tables contains odd spacing, and a lot of white spaces.**
>
> We have tried improving the presentation of the tables and added column lines to the tables.

---

> > ### Comment · Reviewer_ACfY · 2024-11-26
> >
> > Thank you for improving the presentation and addressing my comments. However, I still find MAP as a framework does not substantially differentiate from existing agent framework, therefore I will stick with my original rating of the paper.

---

### Official Review · Reviewer_VMsA · 2024-11-03

**Soundness:** 3
**Presentation:** 3
**Contribution:** 2
**Rating:** 5
**Confidence:** 4

**Summary:**

This paper proposes MAP, a modular, agentic LLM program that uses specialized modules: task decomposer, actor, monitor, predictor, evaluator, and orchestrator, by prompting GPT-4 with in-context learning to perform complex planning tasks. Empirically, the paper shows strong results improving over baselines such as few-shot prompting, multi-agent debate, and tree-of-thought and also shows analysis supporting the need for each component in their pipeline.

**Strengths:**

* The paper is well-written and easy to follow
* The empirical results of the method are compelling, with good performance gains across several planning tasks, and is easy enough to implement from the description in the paper, so it should be readily reproducible and easy to implement

**Weaknesses:**

* Background on planning tasks and assumptions: The authors seem to operate under a fully observable setting that governs how the monitor, predictor, and orchestrator modules are set up. This assumption should be explicitly stated in Section 2, and how it would change when planning in a partially observable setting should be discussed, as well as whether it is in the scope of this work.
* Missing related work: The paper is missing a few relevant works in partially observable settings:
     * [Adapt](https://arxiv.org/abs/2311.05772): This paper also has recursive task decomposition using three modules (which is similar to some of the modules in MAP) for text games that can be argued have planning components in partially observable settings.
     * [LATS](https://arxiv.org/abs/2310.04406): Similarly, this paper also uses several LLM components or modules and fuses them with tree search (MCTS) which is relevant to MAPS. A discussion of the similarities and distinctions between these works would place the work better in the broader context of the existing work.
* Lack of acknowledgment of and addressing computational overhead and cost tradeoff with performance: As noted in Appendix A.8, it can be argued that MAP uses a significantly higher token budget (which also translates into monetary cost) as compared to the baselines to get the performance gains it achieves. A fairer comparison would be to increase the computation budget of baselines such as depth and breadth of tree search and verify that the performance gains persist. Similar to the Adapt paper, the authors can increase the number of tokens generated or report the best of n trials to enable such a comparison.

**Questions:**

* The metrics for evaluating the planning tasks should be discussed explicitly in Sec 3.1
* The presentation of tables 1-3 can be improved by adding the column lines to clearly see the content boundaries and summarize the key takeaways and why you report different baselines for these tasks.
* In Sec 4, it will be helpful to state the numerical margins of improvements in the text for all the plots, since it is not discussed in the paper and is a bit hard to read from the plots.

---

> ### Author Response · Authors · 2024-11-21
> **Author response to Reviewer VMsA**
>
> We thank the reviewer for their helpful suggestions and comments.  Below we provide a point-by-point reply to the concerns raised:
>
> **1. Background on planning tasks and assumptions**
>
> Thank you for this suggestion. We have now added an additional subsection to Section 2 that explicitly states the assumptions of our task setting. This section reads as follows:
>
> 2.1 PROBLEM FORMULATION
>
> We consider planning problems in environments that are both deterministic and fully observable. The environment and transition dependencies are described to the agent, along with a starting state $x$ and a goal $y$, and the agent must generate a plan $P$ (a series of actions) without directly interacting with the environment. Despite the relatively limited nature of the environments we consider (fully observable, deterministic), planning problems of this sort are nevertheless extremely challenging for LLMs (Valmeekam et al., 2023; Momennejad et al., 2023).
>
> We have also added the following note to the section on Future Directions:
>
> Future work should investigate how to extend the proposed approach to more complex
> open-ended environments, especially by incorporating memory mechanisms for
> storing knowledge about the environment as it is accumulated.
>
> **2. Missing related work**
>
> Thank you for bringing these references to our attention. We agree that there are important similarities to our work, namely that Adapt involves task decomposition, and LATS involves tree search. There are several distinctions between these approaches and MAP. One important distinction is that both of these approaches require interaction with the external environment to develop plans, whereas planning is performed internally in our approach. Other distinctions involve the use of predictor and monitor modules which are missing in both of these works. Further Adapt doesn’t use any tree search, and LATS doesn’t decompose a task into subgoals, both of which are present in MAP.  We have added these references to the related work, and have added discussion of the relationship with our approach to the extended related work in Section A.2:
>
> Adapt (Prasad et al., 2023) is another modular agent-based approach that, similar to our
> approach, involves task decomposition, and LATS (Zhou et al., 2023) is another agent-
> based approach that employs tree search. Both of these approaches require interaction with an external environment to develop plans, whereas plans are developed internally in our approach. Furthermore, Adapt doesn’t use any tree search, and LATS doesn’t decompose a task into subgoals, both of which are present in MAP.
>
> **3. Lack of acknowledgment of and addressing computational overhead and cost tradeoff with performance**
>
> Please note that we have included results for an efficient version of MAP that caches and re-uses results for identical prompts (as is also done with tree-of-thought), and this version of the model has a very similar computation budget to tree-of-thought (3.16 per problem for the efficient version of MAP vs. 2.50 per problem for ToT), while substantially outperforming ToT. We have also included metrics showing both the performance of the best plan as well as the average performance for comparing ToT and MAP, and find that MAP outperforms ToT for both metrics.
>
> **4. Metrics for evaluating the planning tasks**
>
> We have now added a subsection that explicitly describes our metrics:
>
> 3.3 METRICS
>
> Our primary metric is the percentage of problems that are solved for any given task. For
> non-deterministic methods (MAP, ToT), we consider both average percent solved (across
> multiple plans per problem) and percent solved for the best plan (according to a
> groundtruth evaluation). For Tower-of-Hanoi and Graph Traversal we also consider the percentage of moves that are invalid (that violates the rules, or attempts to traverse a nonexistent edge).
>
> **5. The presentation of tables 1-3 can be improved by adding the column lines to clearly see the content boundaries and summarize the key takeaways and why you report different baselines for these tasks.**
>
>
> Thank you for this suggestion. We have added column lines to the tables. Due to space constraints, we found it difficult to add these additional details to the table captions, but we would like to note that these details are provided in the text. Specifically, we have highlighted the takeaways from each table in the text, and have also provided additional details on why the ToT baseline is not included in the table for the planbench results (due to the computational cost of performing tree search, but note that ToT results for a subset of problems is included in the appendix Table 13).
>
> **6. In Sec 4, it will be helpful to state the numerical margins of improvements in the text for all the plots, since it is not discussed in the paper and is a bit hard to read from the plots.**
>
> We have now added these details to the text of Section 4.

---

> > ### Comment · Reviewer_VMsA · 2024-11-21
> >
> > Thank you for your efforts and for incorporating my feedback but I will be sticking with my original rating for the paper.

---

### Official Review · Reviewer_6wrt · 2024-11-05

**Soundness:** 3
**Presentation:** 2
**Contribution:** 2
**Rating:** 3
**Confidence:** 4

**Summary:**

The paper introduces the Modular Agentic Planner (MAP), a modular architecture designed to improve the planning capabilities of large language models (LLMs). MAP integrates several specialized modules—such as action proposal, state evaluation, task decomposition, and error monitoring—each fulfilled by an LLM instance to enhance multi-step reasoning and decision-making. These modules work interactively, decomposing complex tasks into subgoals, validating actions, predicting outcomes, and assessing states, which allows the system to adapt and refine its plans dynamically.

**Strengths:**

1. Reasonable Modular Design: The proposed MAP employs a modular architecture that enhances the planning capabilities of large language models through the coordination of dedicated modules for task decomposition, state evaluation, and error monitoring. The design approach is reasonable and systematically structured.
2. Validation Across Multiple Tasks: The authors evaluated MAP on several representative tasks, including graph traversal, Tower of Hanoi, PlanBench, and the multi-step reasoning NLP task StrategyQA, providing a relatively comprehensive demonstration of the method’s applicability.
3. Demonstrated Transfer Performance: MAP shows a certain level of task transferability, suggesting that the approach can adapt to task variations in some cases, offering a reference for applying modular methods to complex tasks.

**Weaknesses:**

1. Lack of Novelty in Agent Design: Numerous studies have already explored agent system design using role-playing, multi-module coordination, or multi-agent frameworks, such as DEPS, JARVIS-1, Voyager, and CAMEL. These approaches have been validated on more complex, open-world benchmarks. For instance, DEPS includes Descriptor, Explainer, Planner, and Selector modules. Compared to these works, what are the specific advantages and unique contributions of this paper?
2. Predictor Module Challenges: The paper introduces a Predictor module, aiming to forecast the next state, which is inherently challenging, particularly in complex and stochastic real-world environments. This essentially functions as a world model, a field still underdeveloped in research. Thus, I believe this approach may struggle to generalize to more complex environments or tasks.
3. Evaluation on Standardized but Limited Tasks: The paper’s evaluation is primarily on two tasks—Graph Traversal and Tower of Hanoi—both of which are well-established problems in graph theory and search, with mature solutions already available. These tasks are not widely recognized as agent benchmarks, and the community may not find the experiments on these tasks compelling.
4. Lack of Comparison with Other Role-Playing Methods: The paper lacks comparisons with other role-playing-based approaches, limiting insight into its relative effectiveness.

[1] DEPS: Describe, Explain, Plan and Select: Interactive Planning with LLMs Enables Open-World Multi-Task Agents
[2] JARVIS-1: Open-World Multi-task Agents with Memory-Augmented Multimodal Language Models
[3] Voyager: An open-ended embodied agent with large language models
[4] CAMEL: Communicative Agents for "Mind" Exploration of Large Language Model Society

**Questions:**

Refer to the weakness, please.

---

> ### Author Response · Authors · 2024-11-21
> **Author response to Reviewer 6wrt**
>
> We thank the reviewer for their helpful suggestions and comments.  Below we provide a point-by-point reply to the concerns raised:
>
> **1. Lack of Novelty in Agent Design**
>
> The novelty of our proposed approach lies in the specific functionality of the modules, and the algorithms through which they interact. Regarding the specific agent-based approaches mentioned here, three out of four of these approaches cannot be applied to the planning problems that we investigate. DEPS cannot be applied to the problems that we investigate because it involves a domain-specific subgoal selection module, based on the proximity to objects in a 2D environment (please note that we describe the relationship with DEPS in the supplementary related work Section A.2). Voyager and JARVIS-1 cannot be applied to the problems that we investigate, because they require interaction with an external environment to iteratively develop plans, whereas we are interested in the more challenging setting in which plans are developed internally.
>
> The fourth approach mentioned here, CAMEL, is a relatively simple framework involving only three modules (a task specifier, that elaborates on a human-generated task prompt, and AI assistant and AI user modules that interact with each other to solve the task). We evaluated a similar approach, multi-agent debate, as a baseline and found that it performed very poorly on our tasks. Furthermore, our ablation experiments showed that each of our proposed modules played an important role in planning, so it is unlikely that this approach will be successful.
>
> Thank you for bringing these works to our attention. We have now referenced these works in the Related Work (section 5) and added discussion of the relationship with our approach to the Supplementary Related Work (section A.2) of the revised manuscript.
>
>
> **2. Predictor Module Challenges**
>
> Thank you for raising this important issue. We agree that it is highly non-trivial to learn a world model, especially in non-deterministic settings, and we do not claim to have solved this problem. However, it is important to note that *even in fully observable and deterministic settings*, where prediction is in principle substantially easier, the planning tasks that we consider are still extremely challenging for LLMs. and for the other agent-based baselines that we consider. To make progress on these issues, in this work we aimed to tackle the planning problem in a setting where prediction is relatively easier, but ongoing research on world models is also important and is indeed synergistic with our work. To clarify this aspect of our work, we have now added a subsection that describes our problem formulation to Section 2:
>
> 2.1 PROBLEM FORMULATION
>
> We consider planning problems in environments that are both deterministic and fully observable. The environment and transition dependencies are described to the agent, along with a starting state $x$ and a goal $y$, and the agent must generate a plan $P$ (a series of actions) without directly interacting with the environment. Despite the relatively limited nature of the environments we consider (fully observable, deterministic), planning problems of this sort are nevertheless extremely challenging for LLMs (Valmeekam et al., 2023; Momennejad et al., 2023).
>
>
> **3. Evaluation on Standardized but Limited Tasks**
>
> Please note that, in addition to our experiments on Graph Traversal and Tower of Hanoi, we have also performed extensive experiments on the PlanBench benchmark (Tables 1,3,11, and 13), which is an extremely challenging planning benchmark for LLMs (e.g., GPT-4 solves only 0.2% of problems in the most challenging setting), and have also evaluated the approach on the StrategyQA benchmark, where it performed on par with the human baseline (Table 2).

---

> > ### Comment · Reviewer_6wrt · 2024-11-25
> >
> > Thank you for your effort and response. However, I will maintain my original evaluation.

---

### Note · Authors · 2024-11-27

I have read and agree with the venue's withdrawal policy on behalf of myself and my co-authors.